# Regulation of the PI3K pathway through a p85α monomer–homodimer equilibrium

Lydia WT Cheung[1], Katarzyna W Walkiewicz[2], Tabot MD Besong[3], Huifang Guo[1], David H Hawke[1], Stefan T Arold[2]*, Gordon B Mills[1]*

[1]Department of Systems Biology, University of Texas MD Anderson Cancer Center, Houston, United States; [2]Computational Bioscience Research Center, Division of Biological and Environmental Sciences and Engineering, King Abdullah University of Science and Technology, Thuwal, Saudi Arabia; [3]Division of Physical Sciences and Engineering, King Abdullah University of Science and Technology, Thuwal, Saudi Arabia

**Abstract** The canonical action of the p85α regulatory subunit of phosphatidylinositol 3-kinase (PI3K) is to associate with the p110α catalytic subunit to allow stimuli-dependent activation of the PI3K pathway. We elucidate a p110α-independent role of homodimerized p85α in the positive regulation of PTEN stability and activity. p110α-free p85α homodimerizes via two intermolecular interactions (SH3:proline-rich region and BH:BH) to selectively bind unphosphorylated activated PTEN. As a consequence, homodimeric but not monomeric p85α suppresses the PI3K pathway by protecting PTEN from E3 ligase WWP2-mediated proteasomal degradation. Further, the p85α homodimer enhances the lipid phosphatase activity and membrane association of PTEN. Strikingly, we identified cancer patient-derived oncogenic p85α mutations that target the homodimerization or PTEN interaction surface. Collectively, our data suggest the equilibrium of p85α monomer–dimers regulates the PI3K pathway and disrupting this equilibrium could lead to disease development.

*For correspondence: stefan. arold@kaust.edu.sa (STA); gmills@mdanderson.org (GBM)

Competing interests: The authors declare that no competing interests exist.

## Introduction

The phosphatidylinositol 3-kinase (PI3K) pathway is a master regulator of many cellular processes. Activation of the class 1A PI3K is gated, in part, by the p85α regulatory subunit, which contains a Src homology 3 (SH3) domain, two proline-rich (PR) regions (PR1 and PR2) separated by a Rho-GAP/BCR-homology (BH) domain, and two Src homology 2 (SH2) domains (nSH2, cSH2) flanking an inter-SH2 (iSH2) domain. The interaction of p85α nSH2-iSH2-cSH2 fragment with the p110 catalytic subunit represses p110 activity allosterically and also stabilizes p110 against degradation, creating a pool of stable but quiescent p110 (*Yu et al., 1998*). Upon receptor tyrosine kinase activation, p85α undergoes phosphorylation/dephosphorylation events that alleviate p110 inhibition resulting in p110-mediated production of PI(3,4,5)P3 (*Cuevas et al., 2001*). Strikingly, *PIK3R1*, the gene coding for p85α, is the twelfth most frequently mutated gene across all cancers. Indeed, somatic mutations of the p85α:p110α interface disrupt the inhibitory action of p85α on bound p110α leading to PI3K pathway activation in cancers (*Miled et al., 2007*; *Jaiswal et al., 2009*). However, *PIK3R1* mutations outside the nSH2-iSH2-cSH2 fragment are relatively common in cancers, particularly endometrial cancer. The mechanisms underlying the transforming activity of these mutations remain to be elucidated and may provide novel biomarkers or therapeutic opportunities.

Although the best-characterized role of p85α is p110-dependent, interaction of p85α with other proteins via its SH3 and BH domains has been proposed to mediate p110-independent functions

**eLife digest** Many cancers arise due to genetic mutations that allow cells to proliferate uncontrollably. Cell proliferation and many other cell processes can be regulated through a signaling pathway that involves an enzyme called PI3K. This 'heterodimeric' enzyme is made up of two protein subunits, one of which is called p85α and inhibits the other subunit of the enzyme (known as p110) to prevent uncontrolled cell proliferation. At the same time, p85α stabilizes p110 and allows the PI3K pathway to be briefly activated when appropriate. Many cancer cells contain mutations in the gene that encodes p85α that prevent the protein from inhibiting p110. This results in the activation of PI3K and promotes cancer formation. A protein called PTEN is a key inhibitor of the PI3K pathway. Common mutations to the *PTEN* gene in cancer cells stop the PTEN protein working efficiently, or prevent PTEN production.

Recent research has revealed that two molecules of p85α that are free from p110 can bind to each other to form a 'homodimer'. Cheung et al. have now used biochemical, cell biological and computational methods to investigate the role of these p85α homodimers. This revealed that p85α homodimers stop PTEN being broken down by binding to it. As a consequence, there is enough PTEN in the cell to inhibit the PI3K pathway.

By examining the mutations present in cancer patients, Cheung et al. next identified mutations that prevent the p85α protein from forming homodimers, or that prevent the homodimers from interacting with PTEN. PTEN therefore degrades and cannot inhibit the PI3K pathway, which allows the cells to proliferate. Methods that increase p85α homodimer formation or enhance the ability of p85α homodimers to bind to PTEN may therefore provide new approaches for developing cancer treatments. More generally, it appears that maintaining the correct balance between the amount of p85α in the form of p110-bound heterodimers and p110-free homodimers in a cell may be important for preventing diseases involving the PI3K pathway.

(*Jimenez et al., 2000*; *Chamberlain et al., 2010*). We and others have also shown that the BH domain binds to the tumor suppressor PTEN to promote PTEN protein stability and that at least one *PIK3R1* somatic mutation interferes with this process (*Chagpar et al., 2010*; *Cheung et al., 2011*). Moreover, intermolecular interactions between the SH3 domain and the PR region of p85α contribute to p85α homodimerization (*Harpur et al., 1999*; *Cheung et al., 2011*). However, the function of p85α homodimers remains to be elucidated. In this study, we demonstrate that p110α-free p85α homodimers positively regulate PTEN and that this regulatory mechanism is disrupted by mutations in a subset of endometrial cancers. Together, our findings suggest that the relative abundance of p110α-bound p85α monomer and p110α-free p85α homodimer is critical in PI3K pathway regulation.

## Results

### The p85α PR1:SH3 domain interaction contributes to stabilization of the p85α homodimer and PTEN binding, whereas PR2 contributes to PTEN binding but not to homodimer formation

Previous analyses suggested that the region encompassing SH3-PR1-BH of p85α mediates p85α homodimerization and binding to PTEN (*Harpur et al., 1999*; *Chagpar et al., 2010*; *Cheung et al., 2011*). However, how each motif contributes to homodimerization and the orientation of the homodimer remain unknown. Using analytical ultracentrifugation (AUC) and microscale thermophoresis (MST), we demonstrated that purified recombinant full-length p85α homodimerized with a micromolar dissociation constant $K_d$ in absence of other proteins under reducing conditions ($K_d$ was $7 \pm 0.7$ µM and $3.9 \pm 0.2$ µM for AUC and MST, respectively) (*Figure 1—figure supplements 1, 2*). AUC and MST also showed that the SH3-PR1-BH-PR2 fragment retained the full capacity to dimerize ($K_d$ was $0.53 \pm 0.03$ µM and $0.44 \pm 0.03$ µM for AUC and MST, respectively). The difference in dimerization $K_d$ between the N-terminal fragment and full-length p85α might indicate an additional entropic penalty arising upon full-length dimerization and/or weak intramolecular interactions

occurring between the SH3-PR1-BH-PR2 and the nSH2-iSH2-cSH2 fragments in the monomer, which have to be displaced to allow dimerization.

The isolated SH3 domain does not form stable dimers in size-exclusion chromatography (*Harpur et al., 1999*) and none of the available p85α SH3 crystal structures ([PDB 3I5S (*Batra-Safferling et al., 2010*), 3I5R (*Batra-Safferling et al., 2010*), 1PHT (*Liang et al., 1996*)] contains quaternary assemblies predicted to be stable in solution by the PISA algorithm (*Krissinel and Henrick, 2007*), suggesting that SH3:SH3 contacts do not have a major role in p85α homodimerization. We therefore asked if the p85α homodimer could be stabilized by SH3:PR1 interactions in trans. p85α PR1 (residues 79–99) contains a canonical class I PXXP SH3-interacting motif ([R/K]XXPXXP; R$^{93}$PLP$^{96}$VAP$^{99}$) (*Ladbury and Arold, 2011*). Indeed, our isothermal titration calorimetry (ITC) analyses showed that the synthetic p85α PR1 peptide PKPRPP**R**PL**P**VA**P** bound purified recombinant p85α SH3 with a $K_d$ of 24 μM (*Figure 1—figure supplement 3M*). p85α SH3 bound p85αΔSH3 (residues 86–742, comprising PR1 to cSH2) with a similar $K_d$ of 17 μM (*Figure 1—figure supplement 3N*), demonstrating that the p85α SH3 domain binds PR1 also in the context of an almost complete p85α, in agreement with previously published qualitative data (*Harpur et al., 1999*). The linker between SH3 and PR1 in p85α is too short for an SH3:PR1 interaction to occur in *cis* (*Figure 1—figure supplement 3O*), indicating that the SH3: PR1 interaction would need to form in *trans* in the p85α homodimer. Two published experimental structures provide comparable templates for this interaction: (i) p85α SH3 in a complex with a class I PXXP peptide that is similar to PR1 (HSK**R**PL**P**PL**P**SL; $K_d$ of 40 μM; [*Batra-Safferling et al., 2010*]) and (ii) p85α PR1 in a complex with Fyn SH3 ($K_d$ 16 μM) (*Renzoni et al., 1996*). We used this structural information to build a theoretical homology model for the SH3:PR1 interaction to guide studies aimed at identifying the amino acids involved in the molecular interactions (*Figure 1A*).

We mutated either the key prolines or basic residues in the PR1 and PR2 regions proposed to bind p85α SH3 (*Figure 1B*). These mutants were expressed in an endometrial cancer cell line KLE, which expresses low level of endogenous p85α and does not have mutations in major members of the PI3K pathway. Co-immunoprecipitation analysis showed that mutations in prolines (denoted as PR1mut) or basic residues (PR1mut') in PR1 decreased p85α homodimerization (*Figure 1C*). Consistent with our hypothesis that homodimerized p85α binds PTEN, PR1mut, and PR1mut' decreased the association of p85α with PTEN (*Figure 1C*). Interestingly, both PR2mut (proline residues mutated) and PR2mut' (basic residues mutated) decreased binding to PTEN without affecting p85α dimerization (*Figure 1C*), suggesting that PR2 contributes directly to PTEN binding rather than to p85α homodimerization. Accordingly, PTEN binding was further decreased by combined mutations of PR1 and PR2 (PR1+ PR2mut and PR1+PR2mut'), which however did not decrease p85α homodimerization compared to PR1 mutations alone. These observations were replicated in p85α knockout mouse embryonic fibroblasts (MEFs) (*Figure 1—figure supplement 4A*) and more importantly in PTEN knockout MEFs (*Figure 1—figure supplement 4B*), indicating that the p85α homodimer is able to form in the absence of PTEN.

We next mutated p85α SH3 residues predicted to mediate binding to PR1 (*Figure 1A*). The D21A/ W55A double mutant, which targets the canonical RXXPXXP binding site, decreased p85α homodimerization and PTEN interaction comparably to PR1mut or PR1+PR2mut (*Figure 1D*). Mutations of E19 and E20, which situate on the SH3 RT loop and may contribute to long-range electrostatic interactions with PR1 (e.g., K88 and R90), also decreased p85α homodimerization and PTEN binding (*Figure 1D*). The D21A/W55A and E19A/E20A mutations further decreased p85α homodimerization and hence PTEN binding when combined with PR1+PR2mut (*Figure 1E*). In the p85α SH3:HSK**R**PL**P**PL**P**SL crystal structure (3I5R) (*Batra-Safferling et al., 2010*) and our p85α SH3:PR1 model (*Figure 1A*), R18 weakly interacts with R93 of the PR1 RXXPXXP motif via a planar stacking interaction. Accordingly, R18A modestly decreased p85α homodimerization (*Figure 1D*).

None of these mutants altered the interaction of p85α with p110α (*Figure 1C–E*, *Figure 1—figure supplement 4A,B*), consistent with C-terminus but not N-terminus of p85α contributing to p110α binding. The spatial separation of p85α homodimerization from p110α interaction sites is further supported by our observations that a truncation p85α mutant, A360*, which lacks the nSH2-iSH2-cSH2 fragment to bind p110α, displayed comparable p85α homodimerization and PTEN binding to wild-type (WT) p85α (*Figure 1—figure supplement 4C*). It is noteworthy that none of the mutations in PR1 disrupted binding to PTEN without affecting p85α homodimerization, consistent with a causal link between homodimerized p85α and PTEN binding. Together, these results suggest that a canonical SH3:PR1 interaction in trans contributes to the formation of the homodimeric p85α platform for

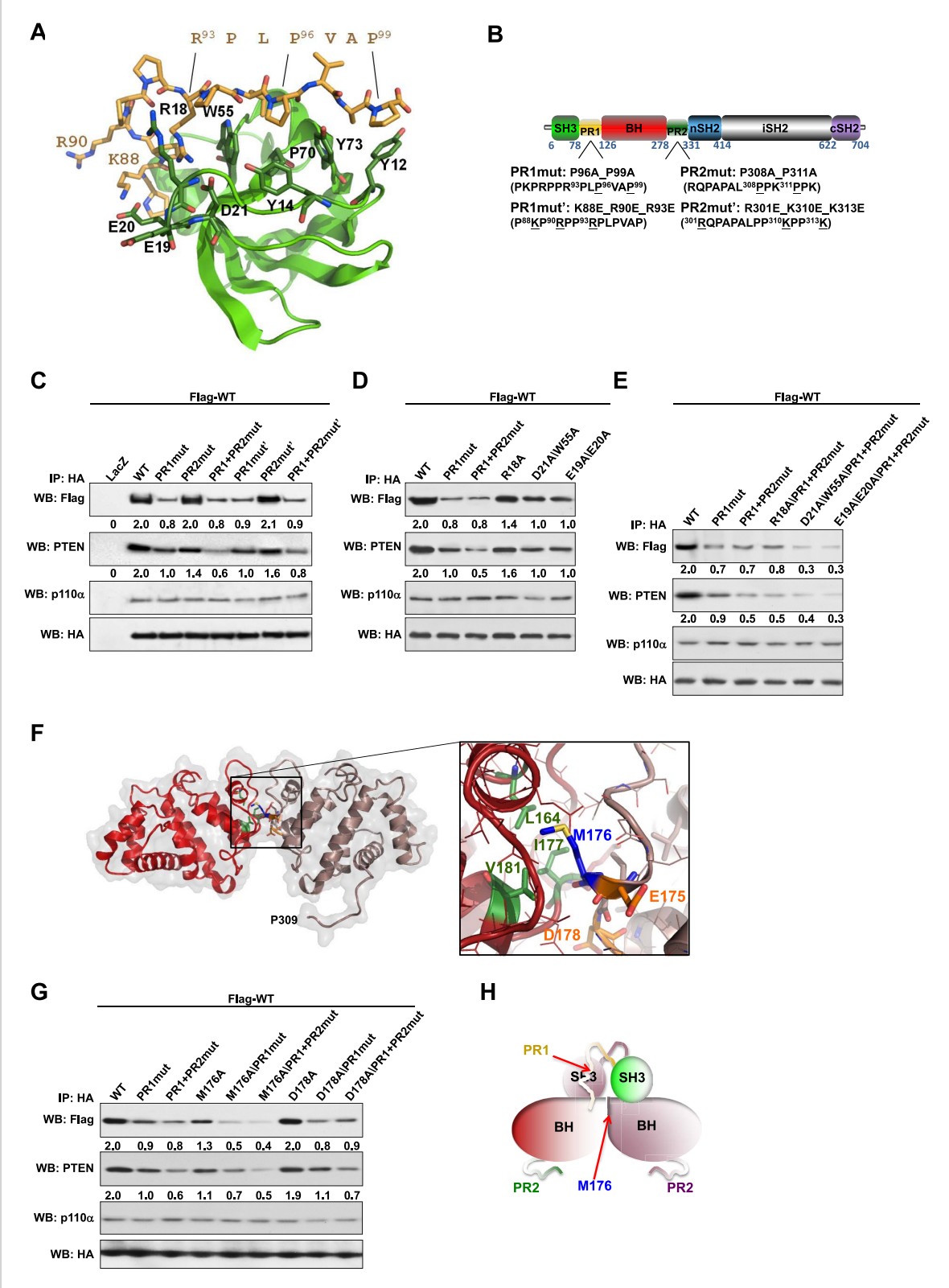

**Figure 1**. Intermolecular interactions contribute to p85α homodimerization and PTEN binding. (**A**) Theoretical molecular model of the p85α SH3:PR1 interaction. The Qualitative Model Energy Analysis (QMEAN) Z-score of the model is −0.8 [the QMEAN Z-score ranges from −4 (worse) to +4 (best) with the average for high-resolution X-ray structures being 0 (*Benkert et al., 2009*)]. Key residues on the Src homology 3 (SH3) domain (green) are highlighted. The RXXPXXP motif of the PR1 region (carbons colored in amber) is indicated. (**B**) Schematic showing the residues mutated in the PR1 and PR2 mutants. *Figure 1. continued on next page*

*Figure 1. Continued*

(**C**) KLE cells co-transfected with Flag-tagged wild-type (WT) p85α (Flag-WT) and HA-tagged WT p85α or proline-rich (PR) mutants for 72 hr were harvested for immunoprecipitation (IP) with anti-HA and Western blotting (WB). (**D**, **E**) In other sets of the experiment, cells co-transfected with Flag-WT and mutants in SH3 domain (**D**) or both SH3 and PR domains (**E**) were harvested for IP using the same Materials and methods. (**F**) The molecular structure of BH:BH domain dimer, taken from the crystal structure of this domain (1PBW). Individual monomers are color-coded. P309, the last PR2 residue modeled in the crystal structure, is indicated. The zoom-in window shows details of the BH:BH interaction, with residues discussed in this study highlighted. (**G**) KLE cells co-transfected with Flag-WT and HA-WT or mutants in BH domain were collected for IP with anti-HA. LacZ was used as control. Numerical values below each lane of the immunoblots represent quantification of the relative protein levels by densitometry (normalized to HA levels). (**H**) Schematic model of the p85α SH3-PR1-BH homodimer. The nSH2-iSH2-cSH2 fragment is not included the model.

The following figure supplements are available for figure 1:

**Figure supplement 1**. (**A–F**) Sedimentation equilibrium data for full-length p85α, p85α PR1-SH3-BH-PR2 (residues 1–333), and p85α PR1-BH (residues 79–301).

**Figure supplement 2**. (**J–L**) Microscale thermophoresis (MST) data on dimerization of p85α (full length, SH3-PR1-BH-PR2 and PR1-BH).

**Figure supplement 3**. (**M–N**) Isothermal titration calorimetry (ITC) data on p85α SH3:PR1 and p85αΔSH3 interactions.

**Figure supplement 4**. (**A, B**) p85α knockout mouse embryonic fibroblast (MEF) cells (A) or PTEN knockout MEF cells (B) co-transfected with Flag-tagged WT p85α (Flag-WT) and HA-tagged WT p85α or PR mutants for 72 hr were collected for IP with anti-HA and WB.

**Figure supplement 5**. (**A**) KLE cells co-transfected with HA-tagged WT p85α (or p85β) and Flag-tagged WT p85α (or p85β) for 72 hr were harvested for IP with anti-HA and WB.

PTEN binding, whereas PR2 contributes to binding to PTEN through an alternative mechanism and potentially by direct PTEN binding.

## The p85α BH:BH domain interaction contributes to stabilization of the p85α homodimer

The p85α BH-PR2 fragment (residues 105–319) forms BH:BH domain dimers in two different crystal forms, suggesting that this dimer arrangement is not an artifact of a particular crystallization condition (*Musacchio et al., 1996*). The crystallographic dimer is stabilized by M176 that inserts into a hydrophobic pocket formed mainly by L164′, I177′, V181′ (*Figure 1F*). The BH:BH interface in the crystal structure is highly conserved in vertebrates. As experimental support for this BH dimer, we found that a M176A mutation decreased p85α homodimerization and PTEN binding (*Figure 1G*). In contrast, mutation of D178, which was predicted to contribute to BH:BH dimer formation only minimally in our model (*Figure 1F*), did not diminish p85α homodimerization.

In each monomer, the BH:BH association buries only 527 Å$^2$ of the total solvent-accessible area of 9000 Å$^2$, suggesting that the BH:BH interaction alone is insufficient for stable p85α homodimerization. Indeed, we measured a much lower affinity for p85α PR1-BH (AUC: 163 μM; MST: 23 μM) than for constructs containing the SH3-PR1-BH region (*Figure 1—figure supplements 1C and 2L*). Together with the decrease in p85α homodimerization that occurred with combined mutations of PR1 and M176 (*Figure 1G*), these data support a model wherein the p85α homodimer is stabilized by both SH3:PR1 and BH:BH interactions. The length of the SH3–BH linker is sufficient for SH3:PR1 interactions in trans within the context of the crystal-derived BH:BH dimer (*Appendix figure 3* in Appendix 1). Collectively, our data are consistent with a molecular model for the p85α homodimer in which the SH3 domain binds PR1 in trans through a canonical class I interaction, while simultaneously the two BH domains associate through an interface centered on M176 (*Figure 1H*). The contribution of the BH domain in the homodimerization of p85 appears to be important, because p85β, which has a completely conserved SH3 ligand-binding site, an almost identical sequence in the PR1 motif and an 80% identity in the nSH2-iSH2-cSH2 fragment, homodimerized and interacted with PTEN to a lesser degree than p85α (*Figure 1—figure supplement 5A*). Indeed, the p85α and p85β BH domains share only 30% identity, and in particular the region that mediates BH dimerization in p85α is not conserved

in p85β, neither in sequence nor in the published crystallographic structure (p85α: PDB 1PBW and p85β: PDB 2XS6) (*Figure 1—figure supplement 5B*).

## Disruption of p85α homodimerization is associated with decreased stability of PTEN through ubiquitination

Our previous data suggested that p85α homodimerization and PTEN binding contribute to increased PTEN stability by inhibiting ubiquitination (*Cheung et al., 2011*). We therefore investigated whether p85α mutants that alter formation of p85α homodimers and/or PTEN binding affect PTEN ubiquitination and stability. WT p85α markedly decreased PTEN ubiquitination compared with LacZ control and more importantly with each of the homodimer-disrupting mutants (R18A, E19A/E20A, D21A/W55A, and R66A in SH3; PR1mut, PR1mut' in PR1; M176A in BH) and PTEN binding-disrupting mutants (PR2mut and PR2mut') (*Figure 2A–C*). Ubiquitinated PTEN in cells expressing the mutants was less than that in cells expressing LacZ because there was more PTEN-bound homodimer present in the mutant cells. PR1+PR2mut and PR1+PR2mut' increased ubiquitinated PTEN compared with single mutations, suggesting cooperativity between PR1 and PR2 in preventing PTEN ubiquitination. Further, the increased PTEN ubiquitination in the presence of these mutants was associated with decreased PTEN protein levels and hence activation of the PI3K pathway as indicated by increased AKT phosphorylation (*Figure 2D–F*). Conversely, p85β failed to stabilize PTEN, as expected from the reduced dimerization and altered BH domain structure of this isoform (*Figure 1—figure supplement 5C*).

A number of feedback and feedforward regulations exist along the PI3K pathway, including negative feedback by S6K, positive feedback by mTORC2, and feedforward activation of mTOR by AKT (*Sekulic et al., 2000*; *Harrington et al., 2004*; *Hahn-Windgassen et al., 2005*; *Sarbassov et al., 2005*; *Humphrey et al., 2013*) (*Figure 2—figure supplement 1A,E,I*). The differences in AKT phosphorylation in presence of mutant or WT p85α raised the possibility that feedback or feedforward signaling mechanisms contributed to the observed changes in PTEN interactions and stability. We therefore examined whether these regulatory mechanisms affect the interaction between PTEN and WT p85α or the PR1+PR2mut. S6K but not Rictor siRNA altered AKT phosphorylation suggesting the existence of S6K-mediated but not mTORC2-mediated feedback in the KLE cell line, although the siRNAs decreased the expression of the proteins by 80% (*Figure 2—figure supplement 1B,C and 1F,G*). Importantly, the S6K and Rictor siRNAs had no effect on the PTEN interactions (*Figure 2—figure supplement 1D,H*). Interference with mTOR or AKT signaling by inhibitors also did not alter the interaction (*Figure 2—figure supplement 1J,K*). These data supported that the observed effects on PTEN stability result from the direct interaction with p85α, rather than from indirect feedback or feedforward-signaling mechanisms.

## p85α homodimer enhances lipid phosphatase activity and membrane association of PTEN

Phosphorylation of the PTEN C-terminal tail residues S380/T382/T383 (hereafter referred to as phosphorylated PTEN) regulates PTEN activity and stability, with the unphosphorylated counterpart (denoted hereafter as unphosphorylated PTEN) being more active (*Vazquez et al., 2000*; *Odriozola et al., 2007*). Consistent with a previous study showing that p85α preferentially binds unphosphorylated PTEN (*Rabinovsky et al., 2009*), we found that p85α increased levels of unphosphorylated PTEN to a greater extent than that of phosphorylated PTEN (*Figure 2D–F*). Intriguingly, p85α also increased PTEN phosphatase activity towards soluble PIP$_3$ substrate in a recombinant cell-free system (*Figure 2G*). This increase in phosphatase activity was recapitulated when PTEN was immunoprecipitated from WT p85α-transfected cell lysates indirectly using anti-p85α antibodies (*Figure 2H*) or directly using anti-PTEN antibodies (*Figure 2—figure supplement 2*). Importantly, the increased phosphatase activity was reversed by mutations in PR1 and PR2 (*Figure 2H* and *Figure 2—figure supplement 2*). The PIP$_3$ substrate of PTEN is located on the cell membrane with unphosphorylated PTEN exhibiting stronger membrane association than phosphorylated PTEN (*Odriozola et al., 2007*). While p85α globally increased PTEN levels in the cytosol, membrane, and nucleus, p85α markedly increased unphosphorylated PTEN, but not phosphorylated PTEN, in the membrane (*Figure 2I*). Moreover, binding of recombinant PTEN to large multilamellar vesicles (LMVs), which represent model membranes used to assess protein-membrane association (*Fukuda et al., 1996*; *Davletov et al., 1998*;

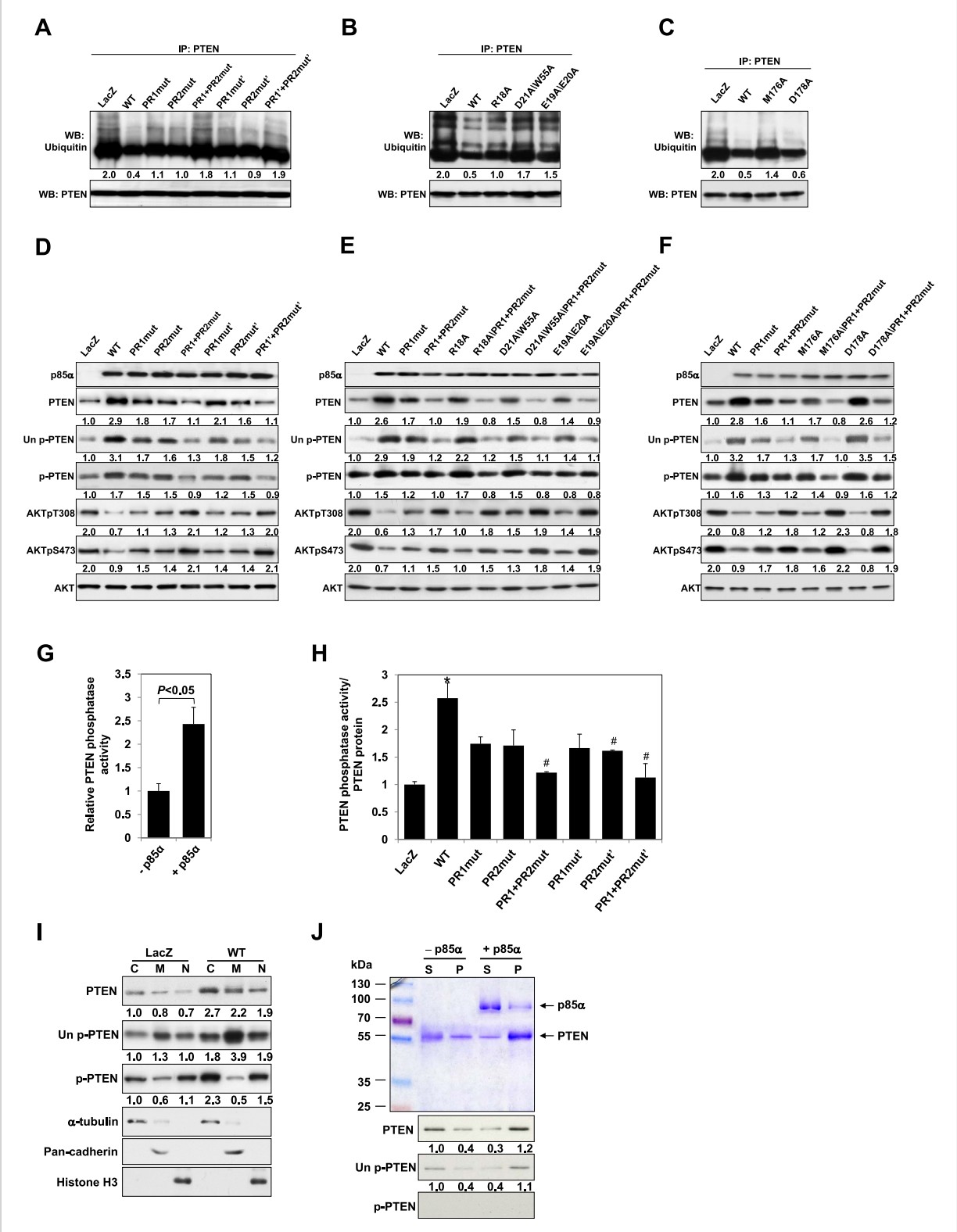

**Figure 2**. p85α homodimer increases protein stability, lipid phosphatase activity, and membrane association of PTEN. (**A–F**) KLE cells transfected with WT p85α or PR1 and PR2 mutants (**A, D**), SH3 domain mutants (**B, E**), or BH domain mutants (**C, F**) for 72 hr were collected for IP with anti-PTEN and WB with anti-ubiquitin (**A–C**) or directly for WB (**D–F**). PTEN protein levels were normalized prior to IP by using proportionally different amounts of lysates. (**G**) In vitro lipid phosphatase activity of recombinant PTEN in the presence or absence of recombinant p85α was determined. (**H**) Endogenous PTEN proteins

*Figure 2. continued on next page*

*Figure 2. Continued*

were immunoprecipitated indirectly using anti-p85α antibody and phosphatase activity was measured. The activity was normalized to the levels of immunoprecipitated PTEN protein in each sample. (I) Transfected KLE cells were harvested for subcellular fractionation and WB (C, cytosol; M, membrane; N, nuclear). (J) Binding of recombinant PTEN to large multilamellar vesicles in the presence or absence of recombinant p85α was assayed. Pellets (P) and supernatants (S) corresponding to phospholipid-bound fraction and phospholipid-unbound fraction, respectively, were subjected to SDS-PAGE followed by top, Coomassie blue staining or bottom, WB. Numerical values below the immunoblots represent relative protein levels by densitometry. *p < 0.05, compared with LacZ control. #p < 0.05, compared with WT. The error bars represent S.D. of triplicates from two independent experiments.

The following figure supplements are available for figure 2:

**Figure supplement 1**. (**A**) Negative feedback of the phosphatidylinositol 3-kinase (PI3K) pathway mediated by S6K.

**Figure supplement 2**. PTEN in vitro lipid phosphatase activity was determined using a malachite green phosphatase assay with soluble PIP3 as the substrate.

*Lee et al., 1999*), was increased in the presence of p85α (*Figure 2J*). Of note, the recombinant PTEN was predominantly in the unphosphorylated form. Together, these data suggest that p85α homodimers likely increase PTEN activity and membrane association of PTEN, in particular unphosphorylated active PTEN.

## p85α homodimer decreases PTEN binding to the E3 ligase WWP2

The ubiquitin-proteasome pathway consists of a cascade of reactions with the substrate specificity being largely defined by E3 ubiquitin ligases. Therefore, we attempted to identify the PTEN E3 ligase that is regulated by the p85α homodimer. Consistent with previous studies (*Wang et al., 2007*; *Maddika et al., 2011*), PTEN interacted with WWP2 and to a lesser extent with NEDD4 (*Figure 3A*). The E3 ligases c-cbl and cbl-b, which bind p85α (*Fang et al., 2001*), did not bind PTEN. Importantly, WT p85α decreased the binding of PTEN to WWP2 without a demonstrable effect on NEDD4 binding (*Figure 3B*). Mutations in PR1 or PR2 increased interactions between PTEN and WWP2, suggesting that homodimerized p85α decreases PTEN ubiquitination by preventing WWP2 from binding PTEN. Again, siRNA and/or inhibitor-based interference with S6K, mTORC2, and AKT signaling had no effect on the interaction of PTEN with WWP2, suggesting that the increase in PTEN:WWP2 association is directly linked to the failure of the p85α mutants to bind PTEN, and not to indirect effects on other PI3K pathway components (*Figure 3—figure supplement 1*). Indeed, both p85α (*Figure 3C*) and WWP2 (*Maddika et al., 2011*) bound to the PTEN phosphatase domain (residues 14–187), suggesting that p85α and WWP2 may compete for PTEN binding. A competition model is supported by the observation that WWP2 dose-dependently decreased the interaction between p85α and PTEN (Left, *Figure 3D*) and p85α-induced PTEN stabilization (Right, *Figure 3D*). Reciprocally, p85α inhibited binding of WWP2 to PTEN (Left, *Figure 3E*) and reversed the decrease in PTEN levels induced by WWP2 (Right, *Figure 3E*). In contrast, p85α PR1+PR2mut (which does not bind PTEN efficiently) failed to reverse the effect of WWP2 (*Figure 3F*). As an additional support of the competition model, siRNAs that target endogenous p85α or WWP2 decreased the interaction of PTEN with WWP2 or p85α, respectively, in an endometrial cancer cell line HEC1A that expresses high levels of p85α and WWP2 (*Figure 3—figure supplement 2*). Together, these data indicate that p85α homodimers compete with WWP2 for binding to an overlapping site on the PTEN phosphatase domain and thereby inhibit PTEN ubiquitination.

## PTEN bound to p85α homodimers does not bind WWP2

To validate this model, analytical gel filtration was performed to characterize the association and competition between p85α, p110α, PTEN, and WWP2. PTEN was recovered in fractions that were free of p110α (*Figure 4A*) and binding of p85α to p110α and to PTEN was mutually exclusive (*Figure 4B*). Coomassie blue staining of p85α immune complex in p110α-containing fractions indicated the existence of other proteins (*Figure 4—figure supplement 1A*) including several known p85α:p110α heterodimer-binding proteins (*Wang et al., 1995*; *Rodriguez-Viciana et al., 1996*; *Lamothe et al., 2004*;

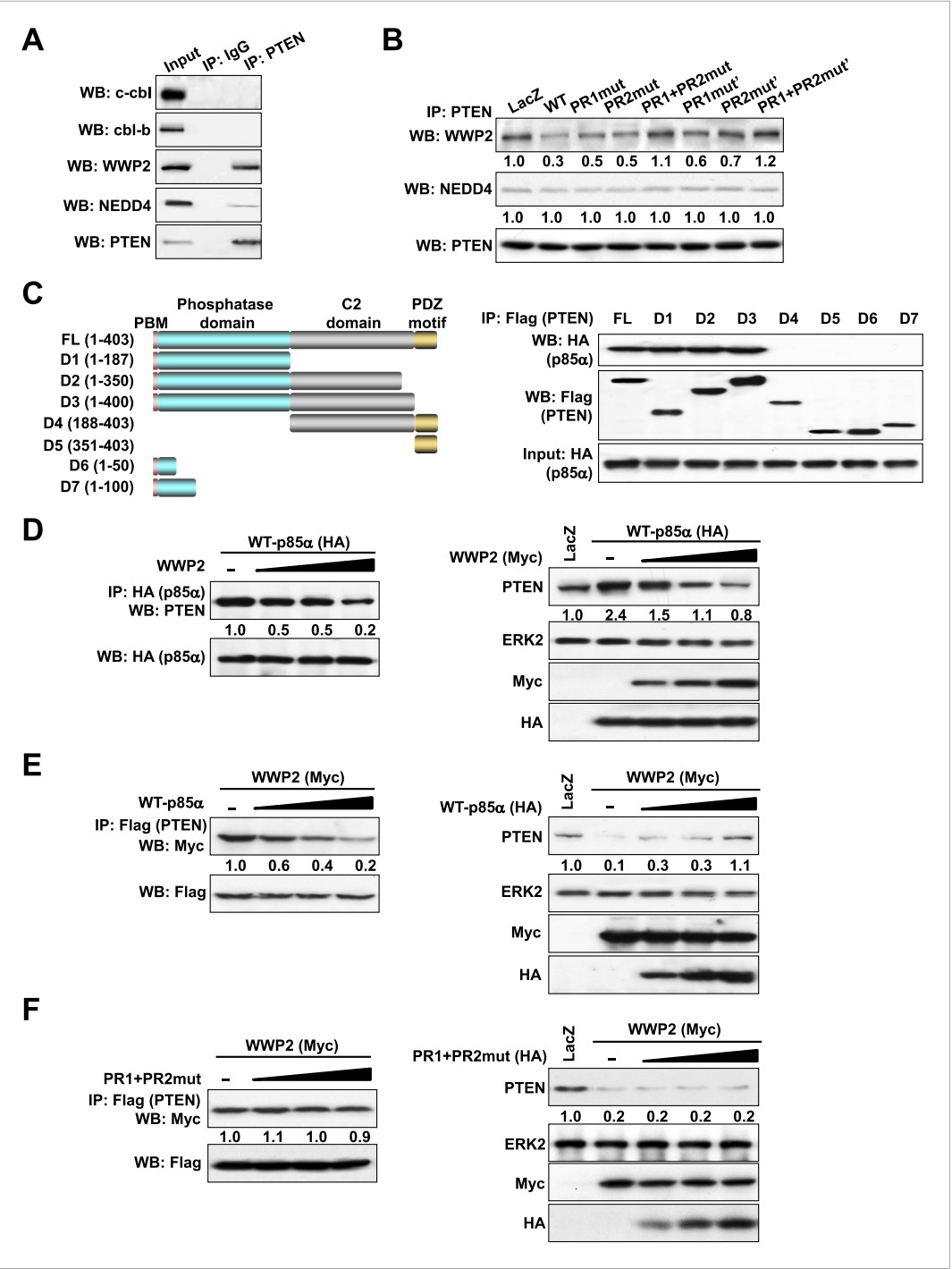

**Figure 3**. p85α homodimer competes with E3 ligase WWP2 for PTEN binding. (**A**) KLE cells were harvested for IP with anti-PTEN and WB. Normal IgG was used as a negative control. (**B**) Cells transfected with WT p85α or PR mutants were harvested for IP after 72 hr. (**C**) Cells were co-transfected with p85α and full-length PTEN (FL) or deletion mutants (Left). PTEN proteins were immunoprecipitated by anti-Flag antibody and the immunoprecipitate was analyzed by WB (Right). (**D**) Cells transfected with HA-tagged p85α in the absence or presence of an increasing amount of WWP2 were collected for IP with HA (Left) or WB (Right). (**E**, **F**) Cells transfected with Myc-tagged WWP2 in the absence or presence of an increasing amount of WT p85α (**E**) or PR mutant (**F**) were collected for IP with Flag for PTEN (Left) or WB (Right). LacZ was used as control. Numerical values below each lane of the immunoblots represent quantification of the relative protein levels by densitometry. PBM, phosphoinositide-binding motif.

*Figure 3. continued on next page*

*Figure 3. Continued*

The following figure supplements are available for figure 3:

**Figure supplement 1**. (**A**, **B**) KLE cells co-transfected with WWP2 and 10 nM siRNA targeting S6K or NS control for 72 hr were harvested for WB directly (**A**) or IP with anti-Flag and WB (**B**).

**Figure supplement 2**. (**A**, **B**) HEC1A cells were transfected with 10 nM siRNA targeting WWP2 (**A**) or p85α (**B**) for 72 hr.

*Asano et al., 2005*) (*Figure 4—figure supplement 1B*). Likewise, consistent with a previous study (*Rabinovsky et al., 2009*), p85α-PTEN existed in a high-molecular weight multi-protein complex (*Figure 4—figure supplement 1A*). The majority of unphosphorylated PTEN eluted in fractions containing high levels of p85α (*Figure 4A*). Immunoprecipitation (IP) confirmed that unphosphorylated PTEN but not phosphorylated PTEN was associated with p85α (*Figure 4B*). Strikingly, no interaction between WWP2 and PTEN was detected in fractions that contained p85α (*Figure 4C*), indicating that WWP2 only binds PTEN that is not bound to p85α in the cell-derived complex.

Next, we determined whether complex formation is altered when p85α homodimerization is disrupted by the PR1+PR2mut. Intriguingly, while distribution of p110α was unchanged, there was a shift in the distribution of unphosphorylated PTEN to fractions lacking p85α (*Figure 4D*), suggesting that the PR mutations decreased the interaction between PTEN and p85α. Moreover, we observed a marked accumulation of PR1+PR2mut in fraction 39 which was absent in WT p85α-transfected cells (*Figure 4A*) that most likely represents monomeric p85α (*Figure 4D*). Importantly, PTEN did not bind PR1+PR2mut in fraction 39 (*Figure 4E–F*). p110α was not detected in fraction 39 likely because p85α was in excess of p110α. Together, these results confirm that p110α binds monomeric p85α, while unphosphorylated PTEN binds homodimeric p85α and further that the binding of PTEN to WWP2 and to homodimeric p85α is mutually exclusive.

## Cancer patient-derived *PIK3R1* mutations target the p85α homodimerization surface

Mutations in *PIK3R1* can contribute to tumorigenesis (*Jaiswal et al., 2009*; *Cheung et al., 2011*). Given p85α homodimerization is a key regulator of PTEN and PI3K pathway activation status, mutations that affect p85α homodimerization or association with PTEN would be expected to increase PI3K pathway activation and tumorigenesis. We therefore searched our in-house endometrial cancer data set (*Cheung et al., 2011*) and The Cancer Genome Atlas (TCGA) (*Figure 5—figure supplement 1*) (*Cerami et al., 2012*) for cancer patient-derived *PIK3R1* mutations that could target p85α homodimerization. Among mutations in the BH domain, E175K (in skin cutaneous melanoma; TCGA) and I177N (in endometrial cancer (*Cheung et al., 2011*)) localize to the BH:BH dimer interface (*Figure 1F*) and occur at sites that are highly conserved across species. I177 is a key residue of the BH:BH hydrophobic core. Substitution of the bulky hydrophobic isoleucine by a smaller and partly polar residue would be predicted to weaken homodimerization. Indeed, I177N decreased homodimer formation, decreased PTEN binding, increased ubiquitinated PTEN, and increased phosphorylated AKT consistent with PI3K pathway activation (*Figure 5A–C*). I177N also promoted interleukin-3 (IL3)-independent survival of the IL3-dependent BaF3 cells (*Figure 5D*) (*Jaiswal et al., 2009*; *Cheung et al., 2011*), consistent with the oncogenic potential of the mutant. These results indicate that I177N could act as an oncogenic mutation at least in part by perturbing p85α homodimerization. Of note, I177N has biochemical phenotypes similar to E160*, which is a patient-derived p85α mutant that disrupts p85α homodimerization and destabilizes PTEN protein (*Cheung et al., 2011*). In contrast, E175 localizes to the polar and charged rim of the BH:BH interface with our model predicting that the E175K mutation would only have minimal influence on the BH:BH interaction (*Figure 1F*). Indeed, biochemical analysis failed to detect effects of E175K on PTEN levels, ubiquitination, p85α binding, and pAKT as well as on proliferation of BaF3 (*Figure 5A–D*).

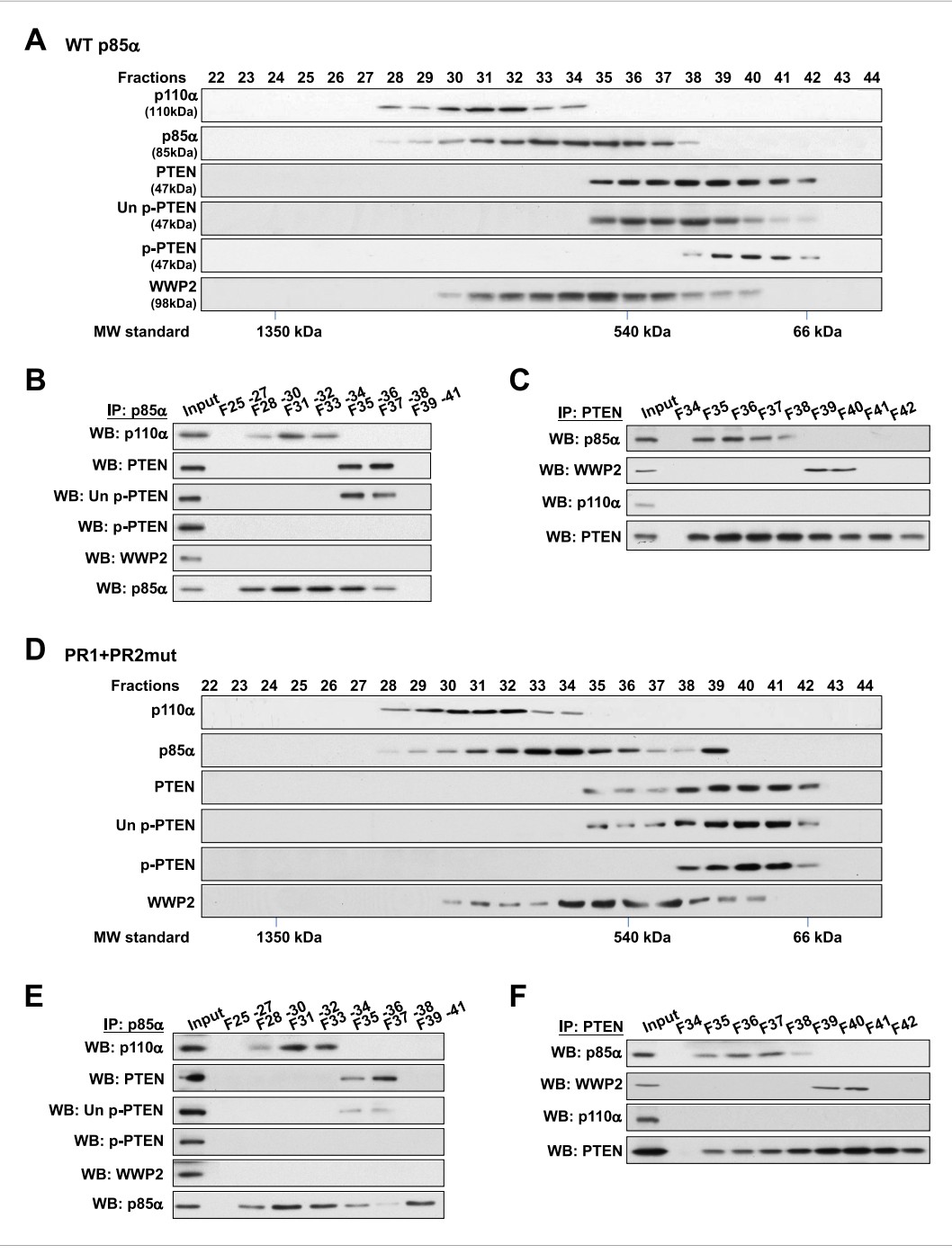

**Figure 4**. Binding of PTEN to WWP2 and to p85α homodimer is mutually exclusive. (**A–F**) Cell lysates from KLE cells transfected with WT p85α (**A–C**) or combined PR1 and PR2 mutant (PR1+PR2) (**D–F**) were fractionated using a gel filtration column and the indicated fractions were analyzed by WB (**A**, **D**) or pooled for IP with anti-p85α antibody (**B**, **E**) or anti-PTEN antibody (**C**, **F**). Input, total lysates before being subjected to gel filtration; F, fraction; MW, molecular weight.

The following figure supplement is available for figure 4:

**Figure supplement 1**. (**A-B**) KLE cells were transfected with WT p85α for 72 hr.

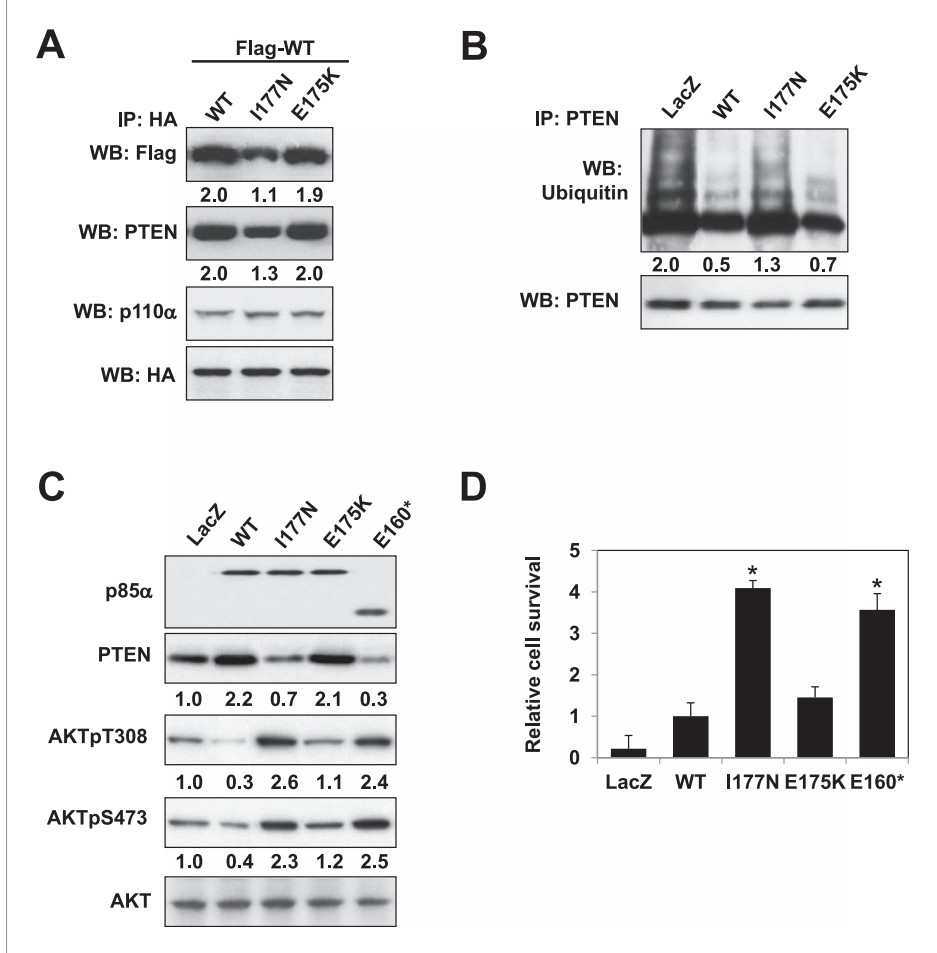

**Figure 5**. Oncogenic cancer patient-derived *PIK3R1* mutation perturbs p85α homodimerization leading to PI3K pathway activation. (**A**) KLE cells co-transfected with Flag-tagged WT p85α (Flag-WT) and HA-tagged WT p85α or patient-derived p85α BH domain mutants were collected for IP with anti-HA and WB. (**B**, **C**) Cells transfected with WT p85α or mutants were collected for IP with anti-PTEN and WB with anti-ubiquitin (**B**) or directly for WB (**C**). PTEN protein levels were normalized prior to IP by using proportionally different amounts of lysates. Numerical values below each lane of the immunoblots represent quantification of the relative protein levels by densitometry. (**D**) Ba/F3 cells transfected with WT p85α or mutants were cultured without interleukin-3 for 4 weeks and harvested for viability assays. *p < 0.05, compared with WT. The error bars represent S.D. of triplicates from three independent experiments.

The following figure supplement is available for figure 5:

**Figure supplement 1**. *PIK3R1* mutations from The Cancer Genome Atlas (TCGA) data sets across tumor lineages are represented by lollipops (green, missense; red, nonsense, frameshift, or splice; black, in-frame deletion/insertion; purple, different types of mutations at the same site).

## Structural modeling of the homodimerized p85α:PTEN complex reveals a cancer patient-derived p85α mutant that is defective at the PTEN interaction interface

Given the pathophysiological relevance of the p85α:PTEN interaction, we next investigated whether the PTEN-interacting interface on p85α would be mutated in cancer patients. To this end, we compiled a total of 16 experimental, bioinformatics, structural and functional constraints (see Appendix 1), including our observation that PTEN and small GTPases bind the p85α BH domain (*Figure 6—figure supplement 1A,B*) non-competitively, to compute the most likely structural model

for the homodimerized p85α:PTEN complex (*Figure 6A* and Appendix 1). In this model, the hydrophobic and charged residues I127/I133/E137 of the p85α BH domain are within the proposed PTEN-binding surface. We therefore engineered a I127A/I133A/E137A triple p85α mutant. The mutant decreased PTEN binding without altering p85α homodimerization (*Figure 6B*). Combined mutations of I127A/I133A/E137A and PR2, which also contributes to interaction with PTEN (*Figure 1C*), resulted in further inhibition of PTEN binding (*Figure 6—figure supplement 1C*), indicating that these residues are cooperative in binding PTEN.

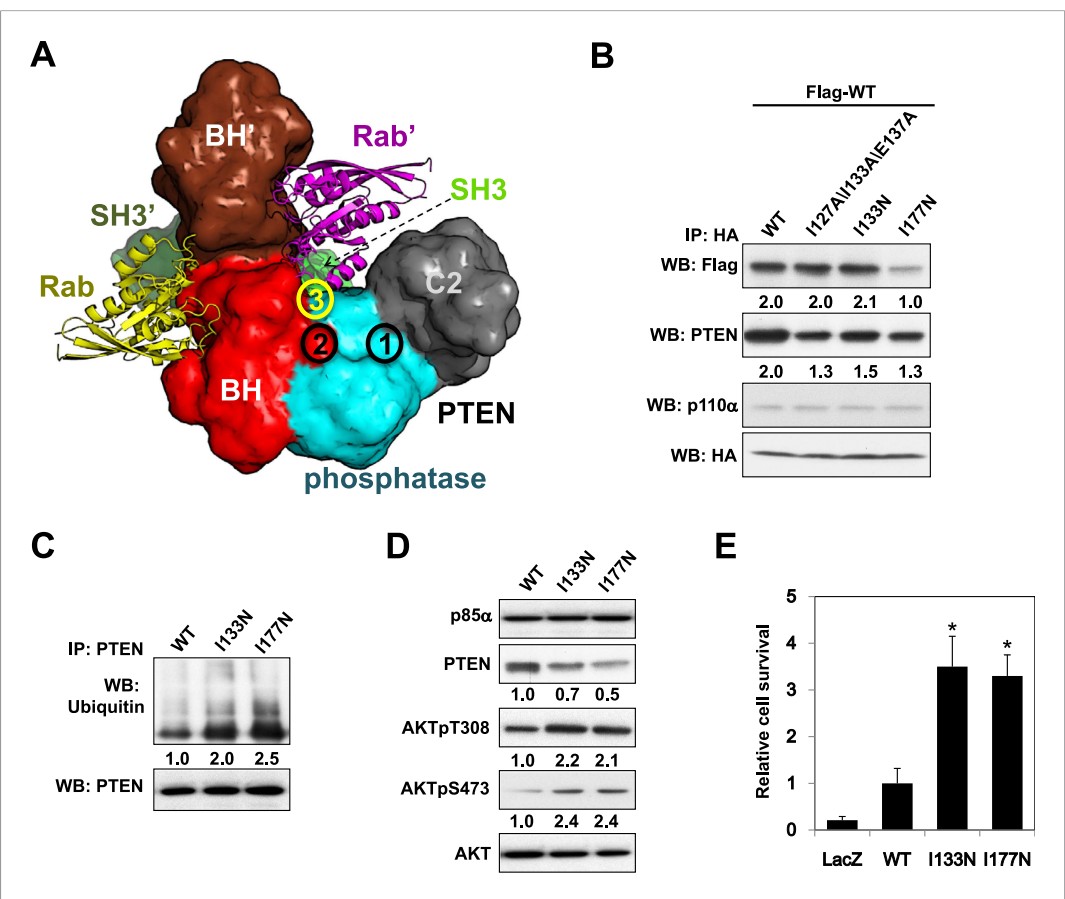

**Figure 6**. Molecular model of the p85α homodimer:PTEN complex reveals cancer patient-derived p85α mutant with decreased PTEN binding. (**A**) Schematic theoretical molecular working model of the homodimerized p85α:PTEN. This speculative model has been constructed by integrating experimental data, physical constraints, and computational scoring functions (see Appendix 1 for details). For simplicity, only one PTEN molecule is shown. The PTEN molecular structure is taken from PDB 1D5R (*Lee et al., 1999*). The model orientation corresponds to the view from the membrane toward the cytosol. Encircled numbers indicate locations of: 1, PTEN phosphatase active site; 2, the side chain of p85α W298, located at the start of the flexible PR2 sequence; 3, approximate position of the PTEN K13 and of the p85α triple mutation I127/I133/E137. (**B**) KLE cells co-transfected with Flag-tagged WT p85α (Flag-WT) and HA-tagged WT p85α or mutants were collected for IP and WB. (**C**) Cells transfected with WT p85α or mutants were harvested for IP and WB. PTEN protein levels were normalized prior to IP by using proportionally different amounts of lysates. (**D**) Cells transfected with WT p85α or mutants were harvested for WB. Numerical values below the immunoblots represent relative protein levels by densitometry. (**E**) Ba/F3 cells transfected with WT p85α or mutants were cultured without interleukin-3 for 4 weeks and harvested for viability assays. *p < 0.05, compared with WT. The error bars represent S.D. of triplicates from three independent experiments.

The following figure supplement is available for figure 6:

**Figure supplement 1**. (**A**) KLE cells co-transfected with Flag-tagged WT p85α (Flag-WT) and HA-tagged WT p85α or mutants were harvested for IP with anti-HA and then subjected to WB.

Strikingly, in an extension of our previous study to additional tumor samples (*Cheung et al., 2011*), we detected a p85α I133N mutant in an endometrial cancer patient. I133N led to a decrease in PTEN binding but not p85α homodimer formation (*Figure 6B*). Comparable to I177N, I133N increased PTEN ubiquitination, induced PI3K pathway activation, and enhanced survival of BaF3 cells (*Figure 6C–E*). Together, these data are consistent with I133N being another oncogenic mutation that blocks stabilization of PTEN. Further, 19 of 62 cancer patient-derived *PTEN* missense mutations in the phosphatase domain mapped to the proposed p85α interaction surface (*Figure 6—figure supplement 1D*), suggesting that additional *PTEN* mutations may function through disrupting the association of PTEN with p85α.

## Discussion

The PI3K pathway is tightly regulated at multiple levels, for example, through PTEN, INPP4B and negative feedback loops such as that mediated by activated S6K (*Harrington et al., 2004*). In this study, we provide a mechanistic model of how the p85α regulatory component of PI3K itself plays a dual role in regulating the PI3K pathway.

A molecular excess of p85 over p110 has been observed (*Ueki et al., 2002*). It has been suggested that p110-free p85 sequesters the adaptor protein insulin receptor substrate (IRS) through an nSH2-iSH2-cSH2 fragment-mediated interaction, thereby competing with the p85:p110 heterodimer for IRS binding and limiting the extent of insulin-induced PI3K signaling (*Luo et al., 2005*). Our results herein suggest an additional but compatible mechanistic model of how free p85α negatively regulates the PI3K pathway.

Our data support a model in which stable p85α homodimerization requires a simultaneous SH3:PR1 interaction in trans and a BH:BH interaction around residue M176. These homodimers can form in absence of other proteins. Collectively, our results illustrate opposing functions of the p85α monomer and homodimer. As a homodimer, p85α competes with WWP2 for binding to the PTEN phosphatase domain and protects PTEN against WWP2-mediated degradation (*Figure 7*). The p85α homodimer also enhances the activity and membrane association of PTEN. Thus, homodimeric p85α indirectly downregulates PI3K signaling. Although the p85α homodimer does not bind p110α, it remains to be ascertained whether the nSH2-iSH2-cSH2 fragment of p85α homodimer can bind phosphotyrosines in activated cell surface molecules and thereby recruit PTEN to the activation nidus. As a monomer, p85α binds p110α in a 1:1 ratio (*Layton et al., 1998*). This interaction is intrinsically inhibitory but it stabilizes p110α and allows activation of the p85α:p110α complex upon stimulation, promoting propagation of ligand-dependent PI3K pathway signaling. We therefore propose that the p85α monomer–dimer equilibrium is a gatekeeper for PI3K pathway activation.

Homodimeric p85α preferentially binds unphosphorylated PTEN, which is less stable yet more active than its phosphorylated counterpart (*Vazquez et al., 2000*; *Odriozola et al., 2007*; *Rabinovsky et al., 2009*). Phosphorylated PTEN is in a 'closed' conformation in which the phosphatase domain is auto-inhibited by the C-terminal tail. In contrast, unphosphorylated PTEN is in an 'open' conformation with the catalytic domain exposed, leading to increased membrane association and activity. The observation that p85α homodimers selectively target the unphosphorylated and less stable form of PTEN fits into a mechanistic model wherein in the absence of excess p85α, and thus, in the absence of p85α homodimers, the unphosphorylated and active PTEN is destabilized, allowing PI3K pathway activation initiated by p110α-bound p85α monomers. Further, our data showed that p85α can bind PTEN, promote PTEN activity and membrane binding in the absence of other proteins. Therefore, the interaction between p85α and PTEN appears to be direct and does not require the presence of other proteins, although p85α and PTEN are part of a high-molecular weight complex containing multiple other proteins.

Based on our data, we speculate that the dynamic formation of p85α homodimers plays a role in the termination of signals initiated at the activation nidus by monomeric p85α-bound p110α. There are several possible scenarios in which the dynamics of p85α homodimer formation could be altered. For example, our data in this and previous studies (*Cheung et al., 2011*) suggested that p110α can disrupt p85α homodimerization by decreasing the abundance of p110α-free p85α. An increase in p110α levels, such as through *PIK3CA* gene amplification, might therefore decrease p85α homodimerization leading to PTEN destabilization. Secondly and intriguingly, *PIK3R1* also encodes for two functional splice variants, p55α and p50α, which lack the domains required for homodimerization. By sequestering p110α, p55α, and p50α could increase the amount of free p85α.

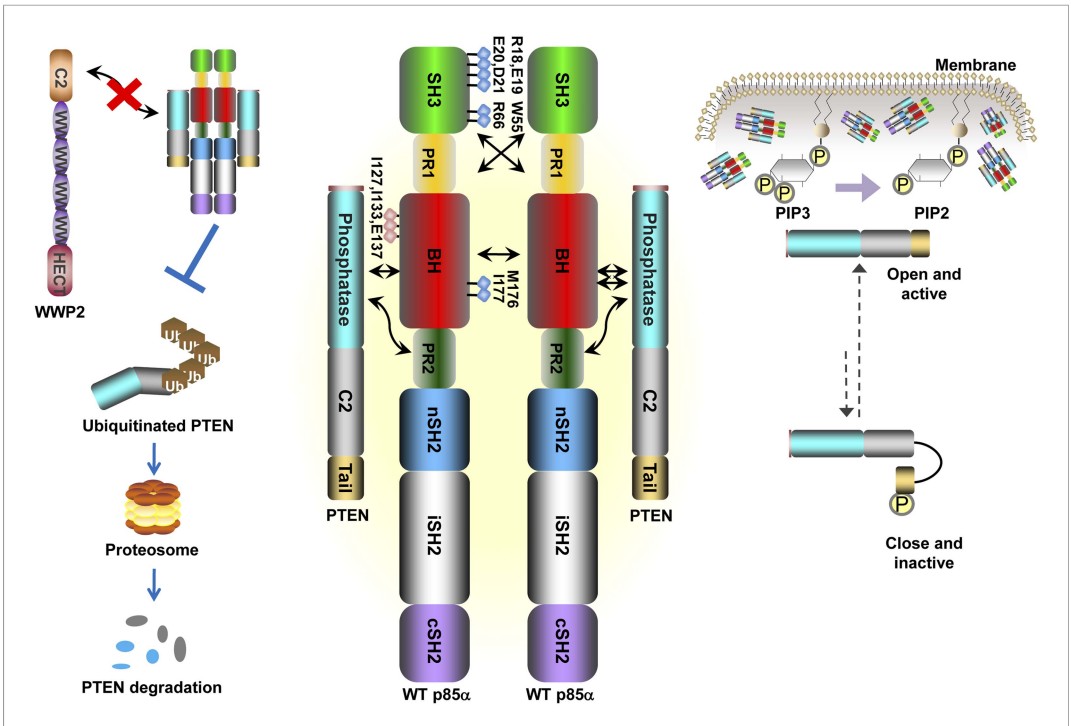

**Figure 7**. Schematic working model of how homodimerized p85α promotes PTEN stabilization and lipid phosphatase activity. Our data support a p85α homodimer model that includes intermolecular interactions between SH3:PR1 in trans and BH:BH interactions between protomers. Key contact residues at the interfaces are shown as blue lollipops. The homodimerized p85α binds PTEN at least partly through the PR2 domain and the indicated residues (red lollipops) in the BH domain. This interaction prevents PTEN from binding to the E3 ligase WWP2, thereby inhibiting PTEN ubiquitination. The homodimerized p85α preferentially binds unphosphorylated PTEN, which has an open conformation and is more active. Independent of stabilizing PTEN protein, homodimerized p85α also enhances PTEN lipid phosphatase activity and association of unphosphorylated PTEN with the membrane where the PIP$_3$ substrates localize. It remains unknown whether homodimerized p85α induces conversion of unphosphorylated PTEN from its close, inactive phosphorylated counterpart.

Moreover, naturally occurring somatic *PIK3R1* mutations (I177N and E160*) and likely others disrupt p85α homodimerization, including mutations at P99 in PR1, R162*, frame shifts at Q153, D168, H180, V181, and L182 (*Cheung et al., 2011*; *Cerami et al., 2012*). Further, a patient-derived p85α mutant I133N that targets the PTEN interaction interface was identified using our speculative theoretical p85α:PTEN model. The *PIK3R1* mutations may hence be selected, at least in part, by effects on p85α homodimer formation and PTEN destabilization. Intriguingly, the proposed p85α-binding site on PTEN also overlaps with a PTEN surface that is frequently mutated in cancers (in 30% of the missense mutations of the phosphatase domain), suggesting that some of these PTEN mutants may disrupt association with p85α therefore decreasing PTEN protein stability and activity.

In addition to predicting mutations that disrupt homodimerization or PTEN interaction interfaces, the homodimerized p85α:PTEN model also offers possible mechanistic explanations for experimental observations. For example, the region outlined by I127/I133/E137 is close to in space to PR2 and both regions could thus together constitute a binding site for PTEN, possibly explaining the cooperativity between these residues in PTEN binding (*Figure 6—figure supplement 1C*). The model also proposes that PTEN:BH interaction renders a site of PTEN ubiquitination K13 inaccessible, suggesting a mechanism for how p85α could protect PTEN from ubiquitination (*Figure 6A*). Moreover, the model predicts that the phosphorylated C-terminal tail of the 'closed' inactive PTEN competes with the p85α PR2 for binding to the phosphatase domain of PTEN, providing a plausible explanation for why p85α selectively binds unphosphorylated PTEN.

One key insight from our efforts to establish a speculative theoretical p85α:PTEN model was that our experimental constraints (derived from our mutational analysis, and from our observation that the PTEN:p85α association is compatible with the p85α:GTPase interaction and with PTEN catalytic activity) cannot be satisfied by a structural model in which one PTEN phosphatase domain binds simultaneously to both BH domains of the p85α homodimer. Instead, our theoretical model proposes that the PTEN phosphatase domain binds to the BH domain of only one p85α protomer, opposite to the GTPase-binding site (*Figure 6A* and Appendix 1). To explain the requirement of a stable p85α homodimer for PTEN binding and protection, our model suggests that (i) although the PTEN phosphatase domain only binds to one BH molecule, the flexible N-terminal 13 residues of PTEN may interact with the second BH molecule of the dimer (in our model, PTEN residue 14 is close to and its main chain orients towards the BH:BH domain interface) and/or (ii) homodimerization allosterically affects the PTEN-binding site on BH (p85α residues I127/I133/E137 are close to the BH:BH homodimerization interface). Although our molecular model is supported by experimental data and showed strong predictive power, it remains hypothetical and we cannot exclude that other structural complexes could satisfy our constraints, or that some of the constraints are wrong. For instance, our model assumes that the PTEN:p85α(:GTPase) assembly is symmetric and homogenous. Yet, there could be other possible scenarios, for example, where only one of two p85α homodimer–bound PTEN molecules is able to reach the membrane with its catalytic site, or where only one p85α molecule of the homodimer binds to a GTPase, allowing a PTEN molecule to bind to both p85α molecules of the dimer's other side. Moreover, the contribution of flexible regions in protein interactions makes the structural modeling of these interactions necessarily more uncertain. Accordingly, uncertainty in our speculative PTEN:p85α complex model arises from the lack of precise information concerning the structure and possible interactions of the flexible regions of PTEN (the first 13 residues) and p85α (the PR2 region, and the loop between residues 123 and 130). Thus, while currently representing a useful working model for guiding and informing experimental analysis, our theoretical PTEN:p85α structure requires confirmation by an experimental structure.

In conclusion, our study suggests that the PI3K pathway activation status can be influenced by the relative levels of p110, p85α, and PTEN. These findings lend new insight to how changes in p110 levels through amplification or *PIK3R1* mutations could lead to PI3K pathway hyperactivation and thus contribute to diseases such as cancer. By decreasing PTEN expression and activity, these non-nSH2-iSH2-cSH2 *PIK3R1* mutations may affect sensitivity to particular targeted therapeutics. Indeed, p110β rather than p110α has been proposed as the primary target in cells with PTEN protein loss (*Jia et al., 2008*). Targeting p85α homodimerization or the p85α:PTEN interaction may represent a new avenue for cancer treatment.

## Materials and methods

### Homology modeling
The complex between p85α SH3 and p85α PR1 peptide P$^{87}$KPRPPRPLPVAP was constructed based on the crystal structures of p85α SH3:HSKRPLPPLPSL (3I5R) and Fyn SH3: PPRPLPVAPGSSKT (1A0N). This structure was refined using the Rosetta energy function as implemented in FlexPepDock (*London et al., 2011*). Construction of the PTEN:p85α molecular interaction model is detailed in the Appendix 1.

### Protein production
The p85α SH3 (residues 1–81) and PR1-BH domains (residues 79–301) were produced in *E. coli* BL21 cells as a 6His-fusion protein using the pET28b expression vector (Novagen). Protein production was induced by 0.4 mM Isopropyl β-D-1-thiogalactopyranoside (IPTG) at 37°C. Cells were lysed using mild sonication, and proteins were purified to homogeneity by immobilized nickel affinity column using standard procedures. Eluted protein was dialyzed into 20 mM Tris pH 8.0, 150 mM NaCl, 2 mM EDTA, 5 mM DTT and applied to S75 (Pharmacia) size-exclusion chromatography. Fractions containing pure SH3 were pooled and concentrated.

The p85α SH3-PR1-BH-PR2 fragment (residues 1–333) and WT p85α were produced in E. *coli* BL21 cells as GST-fusion proteins using a pGex6P-2 expression vector (GE Healthcare, Pittsburgh, PA). Protein expression was induced by 0.4 mM IPTG at 18°C overnight, and proteins were purified by GST affinity column. Eluted proteins were dialyzed into 20 mM Tris pH 8.0, 150 mM NaCl, 5 mM DTT. WT

p85α was further purified using MonoQ 10/100 column and eluted by a linear NaCl gradient (20 mM Tris pH 8.0, 1M NaCl, 5 mM DTT). Finally, p85α$_{1-333}$ C147S and WT p85α were applied to a Superdex200 16/600 (GE Healthcare) size-exclusion chromatography. Fractions containing pure proteins were pooled and concentrated.

## MST

Proteins were fluorescently labeled with Alexa[647] according to the manufacture's protocol (L001, Nanotemper technologies, Germany). Labeled proteins were kept at 20 nM. The unlabeled protein was titrated starting at 50,000 nM and serially diluted in 1:1 ratio in reaction buffer (20 mM sodium phosphate pH 8, 150 mM NaCl, 2 mM EDTA, 5 mM DTT). The measurements were performed at 40% LED and 20, 40, and 60% MST power on NanoTemper Monolith NT.115. Data were analyzed using NT Analysis Software (Nanotemper technologies, Germany).

## ITC

p85α SH3 and GST-ΔSH3 p85α were dialyzed in degassed ITC buffer (10 mM sodium phosphate pH 7.5, 150 mM NaCl, 5 mM DTT). Titrations for p85α SH3:PR1 peptide interaction were carried out on a MicroCal ITC200 at 25°C by serially injecting 2 μl of peptide P$^{87}$KPRPPRPLPVAP into the measurement cell that contained p85α SH3 at a concentration of 100 μM. GST-ΔSH3 p85α and p85α SH3 titrations were carried out similarly, with SH3 in the syringe (400 μM) and ΔSH3 p85α in the cell. Ligand concentrations were 10× the cell concentration. Titrations were analyzed using Origin software.

## Sedimentation velocity

The experiments were conducted on a Beckman Coulter XL-A analytical ultracentrifuge at 20°C using absorption optics. Samples were loaded at 17, 9, 5, 3, and 0.7 μM in a standard cell (400 μl). Data were acquired (120 scans) in intensity mode at 280 nm at 6-min intervals with a rotor speed of 48 krpm for p85α PR1-BH, 45 krpm for p85α SH3-PR1-BH-PR1, and 42 krpm for WT p85α. Data were analyzed in SEDFIT 14.3e (Schuck, 2000) using the continuous c(s) distribution model. Solution densities ρ and viscosities η were calculated using the program SEDNTERP 1.09 (Cole et al., 2008). Analyses of sedimentation coefficients were carried out using s range of 0.05–10 with linear resolution of 100 and confidence levels of 0.95. Fits were achieved with root mean square deviations ranging from 0.0030 to 0.0055 absorbance units. Sedimentation coefficients were corrected to standard conditions at 20°C in water, $s^{o}_{20,w}$ using SEDNTERP 1.09.

## Sedimentation equilibrium

The experiments were performed on a Beckman Optima XL-A analytical ultracentrifuge at 20°C. Samples were loaded at 1.0, 0.5, and 0.25 mg/ml into a 2-channel, 12-mm path-length cells (100 μl). Data were acquired at 11.8, 12.8, 13.8 krpm, as an average of 5 absorbance measurements at 280 nm using a radial spacing of 0.001 cm. Experiments were started from zero to the lowest rotor speed by taking scans at 2-hr intervals and testing for equilibrium by determining the differences between successive scans in SEDFIT. The rotor speed was then increased and the process repeated at each rotor speed studied and sedimentation equilibrium was achieved within 48 hr. Data for the individual proteins collected at different speeds and loading concentrations at 280 nm were sorted and assembled in SEDFIT and analyzed globally in terms of monomer–dimer–tetramer (monomer-m-mer-n-mer ) or monomer–dimer model in SEDPHAT 10.58e (Schuck, 2003) with the implementation of mass conservation. The error analysis was performed with 500 (1000 for p85α PR1-BH) Monte Carlo iterations at the 95% confidence level.

## Cell lines, plasmids, and transfection

The KLE cell line was provided by Dr. Russell Broaddus (M.D. Anderson Cancer Center). The p85α knockout MEFs were obtained from Dr Lewis C. Cantley's lab (Weill Cornell Medical College, New York, NY). The *PIK3R1* cDNA has been described (Cheung et al., 2011). Specific mutations were generated using QuikChange Lightning Site-Directed Mutagenesis Kit (Agilent Technology, Santa Clara, CA). Transfection of plasmids was performed using Lipofectamine 2000 (Invitrogen, Carlsbad, CA). All ON-TARGET plus siRNAs and control siRNA were obtained from Dharmacon (Lafayette, CO) and introduced into the cells using Lipofectamine RNAiMAX (Invitrogen) according to the

manufacturer's instructions. We utilized two independent siRNA sequences per target. The sequences of the siRNAs are p85α-CCAACAACGGUAUGAAUAA and UAUUGAAGCUGUAGGGAAA; WWP2-AGACACGUCCGUUGGGCAG and GCUUCACCCUCCCUUUCUA; S6K- CAUGGAACAUUGUGAGAAA and GGAAUGGGCAUAAGUUGUA; Rictor- GACACAAGCACUUCGAUUA and GAAGAUUUAUUGAGU CCUA.

## IP and Western blotting

Whole cell lysates (25 µg) extracted with radioimmunoprecipitation assay buffer (RIPA) lysis buffer were loaded onto sodium dodecyl sulfate polyacrylamide gel electrophoresis (SDS-PAGE). Primary antibodies specific to PTEN, p110α, total Akt, phospho-Akt (Thr[308] or Ser[473]) and total p110α (Cell Signaling Technology, Danvers, MA), HA (Covance, Princeton, NJ), Myc and FLAG (Sigma–Aldrich, St. Louis, MO), ubiquitin (Enzo Life Sciences, Farmingdale, NY) were used. For IP, cell lysates (1 mg) were immunoprecipitated with antibodies against HA or Flag (1 µg) or PTEN (1:500) overnight at 4°C. The immune complexes were collected by incubation with protein A/G agarose (Santa Cruz) for 4 hr before being resolved by SDS-PAGE.

## Gel filtration

Cells were lysed in a hypotonic lysis buffer containing 10 mM HEPES (pH 7.9), 1.5 mM $MgCl_2$, 10 mM KCl, 1 M EDTA, 0.1% NP-40 and protease inhibitors. Lysates (5 mg) clarified by ultracentrifugation (30,000×$g$, 45 min, 4°C) were applied to Superose 6, 10/300 GL column (GE Healthcare) run at 4°C in binding buffer (0.1% NP-40) on a BioLogic HR system (BioRad, Hercules, CA). Elution was performed at 0.1 ml/min, and 0.5-ml fractions were collected. Fractions were analyzed by Western blotting (WB) or pooled for IP with anti-p85α (1:50) (Abcam, Cambridge, MA) or PTEN (1:500) antibody.

## Ba/F3 survival assay

The assay was described previously (*Cheung et al., 2011*). In brief, Ba/F3 cells transfected with WT *PIK3R1* or mutants were cultured in medium without IL-3 for 4 weeks. Cell viability was evaluated using PrestoBlue (Invitrogen) for mitochondrial dehydrogenase activity.

## Lipid phosphatase assay

In vitro PTEN lipid phosphatase activity was determined using malachite green phosphatase assay kit (Echelon Biosciences, Inc., Salt Lake City, UT). Briefly, protein lysates (1 mg) were subjected to IP of endogenous PTEN using anti-PTEN or anti-p85α antibody and the bead complex was resuspended in PTEN reaction buffer before $PIP_3$ substrate was added to initiate the reaction. Reaction mixture with $PIP_3$ but not lysate was used for background correction. The reaction was stopped after 4-hr incubation at 37°C and the supernatant was separated from the beads for activity detection. The beads were used for WB to quantify immunoprecipitated PTEN. The activity was normalized to immunoprecipitated PTEN level in each sample. To assess the activity of recombinant PTEN, 1 µg of PTEN was incubated with 1 µg of recombinant p85α in buffer containing 50 mM Tris–HCl pH 8.0, 50 mM NaCl, and 10 mM $MgCl_2$ (PTEN reaction buffer without DTT) at room temperature for 1 hr before being subjected to the reaction.

## Phospholipid binding assay

The assay was performed in reference to previous studies (*Fukuda et al., 1996*; *Davletov et al., 1998*; *Lee et al., 1999*). Formation of LMVs composing of brain phosphatidylserine (PS), phosphatidyleth-anolamine (PE), and phosphatidylcholine (PC) (molar ratio 40:10:50) resuspended in buffer containing 50 mM Tris-Cl, 150 mM NaCl, 10 mM DTT, 0.001% Triton X-100 (pH 8.0) was obtained from Avanti Polar Lipids, Inc. (Alabaster, Al). PTEN recombinant protein (5 µg) was incubated in 300 µl of buffer containing 50 mM Tris-Cl, 150 mM NaCl, and 0.001% Triton X-100 (pH 8.0) in the presence or absence of 5 µg of recombinant p85α for 1 hr at 25°C. LMVs (150 µg) were then added and the mixture was incubated for another 15 min at 25°C. After centrifugation at 12,000×$g$ for 10 min, the phospholipid pellets were dissolved in SDS sample buffer. The proteins in the supernatants corresponding to the lipid-unbound population were precipitated with 20% trichloroacetic acid and the precipitates were dissolved in SDS sample buffer. Equal amounts of samples from the supernatants and pellets were analyzed by SDS-PAGE and Coomassie Brilliant Blue R-250 staining.

## Statistical analysis

All experiments were independently repeated at least twice, and data are presented as mean values ± SD. The significance of differences was analyzed by a Student's t test. Significance was accepted at the 0.05 level of probability ($p < 0.05$).

## Acknowledgements

We thank the KAUST Bioscience core lab for instrument access, support for recombinant protein production, and experiments with the MicroCal ITC200 calorimeter. We are grateful to Dr Jonathan M Backer (Albert Einstein College of Medicine, NY) for the plasmids and proteins; Drs Subbareddy Maddika (Centre for DNA Fingerprinting and Diagnostics, Nampally, India) and Junjie Chen (MD Anderson Cancer Center) for the WWP2 and PTEN constructs. Our appreciation also goes Dr Osman Bakr (Functional Nanomaterials Laboratory, KAUST) for granting us access to the analytical ultracentrifugation facility. We thank Virginia A Unkefer for editorial help. This work was supported by Uterine SPORE (NCI 2P50 CA098258-06), Cancer Target Discovery and Development grant (NCI U01 CA168394), Stand Up to Cancer/American Association for Cancer Research Dream Team Translational Cancer Research Grant (SU2C-AACR-DT0209), Program Project Grant (5 P01 CA099031) to GBM; and CCSG functional proteomics core (NCI#CA16672) in MDACC; Uterine SPORE career development grant (NCI P50CA098258) to LWT. Research reported in this publication was supported by the King Abdullah University of Science and Technology.

## Additional information

### Funding

| Funder | Grant reference | Author |
|---|---|---|
| National Cancer Institute (NCI) | Uterine SPORE, NCI 2P50 CA098258-06 | Gordon B Mills |
| National Cancer Institute (NCI) | Cancer Target Discovery and Development grant, NCI U01 CA168394 | Gordon B Mills |
| National Cancer Institute (NCI) | Uterine SPORE career development grant, NCI P50CA098258 | Lydia WT Cheung |

The funders had no role in study design, data collection and interpretation, or the decision to submit the work for publication.

### Author contributions

LWTC, STA, GBM, Conception and design, Acquisition of data, Analysis and interpretation of data, Drafting or revising the article; KWW, Acquisition of data, Analysis and interpretation of data, Drafting or revising the article; TMDB, HG, DHH, Acquisition of data, Drafting or revising the article

## Additional files

### Major datasets

The following previously published datasets were used:

| Author(s) | Year | Dataset title | Dataset ID and/or URL | Database, license, and accessibility information |
|---|---|---|---|---|
| Cheung LW, Hennessy BT, Li J, Yu S, Myers AP, Djordjevic B, Lu Y, Stemke-Hale K, Dyer MD, Zhang F, Ju Z, Cantley LC, Scherer SE, Liang H, Lu KH, Broaddus RR, Mills GB | 2011 | High Frequency of PIK3R1 and PIK3R2 Mutations in Endometrial Cancer Elucidates a Novel Mechanism for Regulation of PTEN Protein Stability | http://cancerdiscovery.aacrjournals.org/content/1/2/170.long | Publicly available at American Association for Cancer Research Journals. |

| Author(s) | Year | Dataset title | Dataset ID and/or URL | Database, license, and accessibility information |
|---|---|---|---|---|
| Cerami E, Gao J, Dogrusoz U, Gross BE, Sumer SO, Aksoy BA, Jacobsen A, Byrne CJ, Heuer ML, Larsson E, Antipin Y, Reva B, Goldberg AP, Sander C, Schultz N | 2012 | The cBio cancer genomics portal: an open platform for exploring multidimensional cancer genomics data | http://cancerdiscovery.aacrjournals.org/content/2/5/401.long | Publicly available at American Association for Cancer Research Journals. |

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

## Appendix 1

# Supplemental material on computational modeling of homodimerized p85α:PTEN complex

In the following, we describe how we combined all available data to establish the most likely theoretical working model for the p85α homodimer:PTEN complex. We note that although the p85α:PTEN model is based on experimental constraints and supported by mutational analysis, the model remains speculative.

## A) Establishing a molecular model for the SH3-BH fragment

### (i) Establishing the dimeric BH domain model

The molecular basis for the dimeric BH domain is derived from the C2-symmetric dimer in the crystal structure of p85α residues 105–320 (PDB accession 1PBW; only residues 115–309 are modeled in the crystal structure). We validated the crystallographic BH dimer model by mutational analysis (see main text and *Figure 1G*).

### (ii) Construction of the molecular model for GTPase binding to p85α

The p85α BH domain binds small GTPases (Rab4, Rab5, Cdc42, Rac1) and functions as a GTPase-activating protein (GAP) (*Zheng et al., 1994*; *Chamberlain et al., 2004*). We found that mutations in the GTPases binding site of p85α did not affect PTEN binding and that Cdc42 and Rab5 did not compete with PTEN for binding to p85α (*Figure 6—figure supplement 1A-B*). The interaction between p85α and small GTPases can be modeled based on the available crystal structure of Cdc42 bound to Cdc42GAP (1GRN). Cdc42GAP is 22% identical to p85α BH. The structure of Cdc42-bound Cdc42GAP was superimposed onto the structure of p85α BH (rmsd: 2.48 Å; *Appendix figure 1*). Using the Cdc42:Cdc42GAP positioning from this superimposition, Cdc42 was transferred to p85α BH to produce a BH:Cdc42 model. The BH:Cdc42 model produced no important clashes and put the BH arginine 151 into the position of the Cdc42GAP R305, an important residue for GAP function and interaction, thus confirming the positioning of the GTPase on BH, at least to a domain-resolution precision. We then used the structure of this BH:Cdc42 model to produce a BH:Rab5 model by using bound Cdc42 as a template for bound Rab5 (25% seq. identity). As an alternative approach, we superimposed the crystal structure of GTP analogue-bound Rab5 (1R2Q) onto GAP-bound Cdc42. Both Rab5 models were highly similar (rmsd = 1.12 Å). All obtained models of BH:GTPase are sufficiently structurally compatible and plausible (shape complementarity without steric hindrance) to serve our purpose of delineating the approximate region on BH occupied by a biologically relevant GTPase.

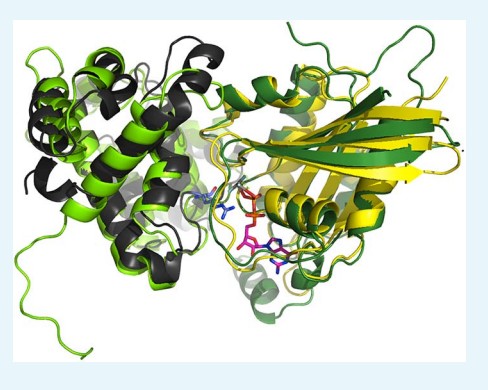

**Appendix figure 1**. Superimposition of the crystallographic complex Cdc42GAP (light green) bound to Cdc42 (dark green) (1GRN) onto p85α BH (black) and the crystal structure of Rab5 (yellow; 1R2Q). The GTP analogue (magenta) and the important catalytic arginine residue (blue) are highlighted.

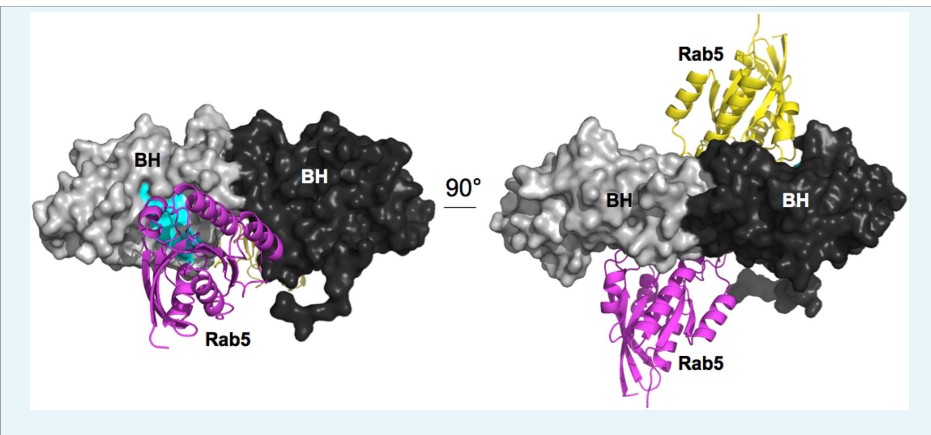

**Appendix figure 2**. Model for Rab5 bound to the BH domain. Left and right panel are 90° views. Cyan: Residues mutated in the GTPase-binding site that did not affect PTEN binding.

We next tested if GTPase binding is stereochemically compatible with formation of the BH dimer. BH:Rab5 did not show any clashes in the context of dimeric BH. Cdc42 showed minor clashes in the region between 121 and 136, but this region is a flexible loop–helix–loop region, with above-average B-factors in the crystal structure, and is expected to adjust to the BH:BH dimer surface. However, to avoid any additional ambiguity due to structural rearrangements, we used Rab5 as the GTPase model in our subsequent analysis.

As seen in the resulting 2x(BH:Rab5) complex (*Appendix figure 2*), compatibility of GTPase binding to BH with PTEN interaction results in strong spatial constraints for both the placement of PTEN and the placement of the SH3:PR1 interaction with respect to the BH domain.

## (iii) Positioning of the SH3:PR1 interaction relative to the BH:BH dimer

Our results strongly suggested that the p85α SH3 domains interact with the PR1 region in trans with a canonical class I PXXP interaction (*Figure 1A*). To ascertain that this type of interaction is physically compatible with the crystallographic BH dimer, we used the program BUNCH (*Konarev et al., 2006*) as a fast and convenient tool, because BUNCH produces stereo-chemically plausible structures, and can include symmetry and contact constraints. Because BUNCH needs a SAXS curve as input to run, we simply simulated a large array of SAXS data sets in order to run BUNCH. We used ten different simulated data sets (calculated based on models with different SH3 positions with respect to the fixed BH domain dimer) to evaluate the impact of the model used to simulate the SAXS data. For each data set, we ran BUNCH 20 times (BUNCH uses Monte Carlo and simulated annealing and hence can give different models for each run, based on the same data). We found that the impact of the model used for simulating data was small compared with the variations in models obtained using different BUNCH runs. In these models, distance constraints were imposed so that the residues of the PR1 RXXPXXP motif were positioned at the right distance with respect to the residues on SH3 with which they interact in our molecular model (*Figure 1A*). Additionally, we augmented the weight of the clash penalty in BUNCH. Using these constraints, BUNCH produced stereochemically plausible solutions where the SH3 domains were positioned either close to (or in contact with) the BH domain of the same molecule (example shown in *Figure 6A*) or of the other protomer; *Appendix figure 3A–C*). Moreover, the SH3 domains were either positioned towards the N-terminus (*Appendix figure 3A,B*), or towards the C-terminus of the BH domains (*Appendix figure 3C*). However, the C-terminal positioning of SH3 domains was incompatible with the interaction of GTPases with the BH domain, because the SH3-BH linker region necessarily crossed the GTPase-binding site on BH, and/or the SH3 domains clashed with the GTPase domains. Given that GTPases and PTEN can bind simultaneously, and given that PTEN binding requires the SH3:PR1 interaction, the C-terminal positioning of SH3 domains was not supported by our data. We therefore concluded that the SH3:PR1 interaction is possible within

the BH:BH domain dimer framework and that the SH3 domain needs to be located at the N-terminal side of the BH:BH domain dimer.

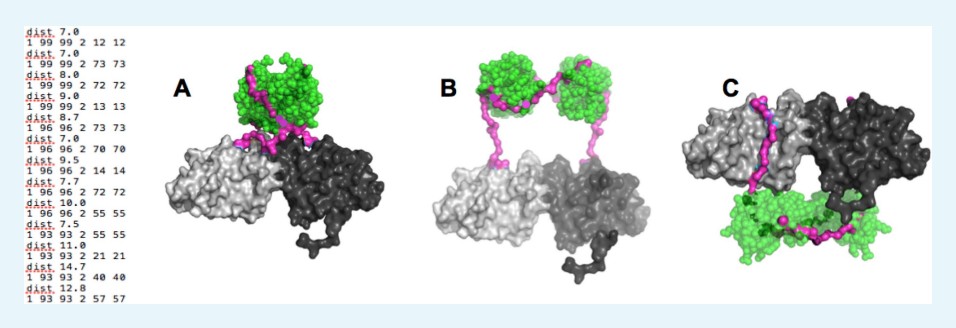

**Appendix figure 3**. (Left) distance constraints used in BUNCH. (**A-C**) Examples of dimeric SH3-PR1-BH models obtained by BUNCH. SH3: green; SH3-BH linker: magenta; BH dimer: light and dark gray. DOI: 10.7554/eLife.06866.024

## B) Molecular model for the PTEN:p85α interaction

Our experimental analysis showed that PTEN is active when in complex with p85α. Moreover, we showed that this interaction strongly depends on both PR2 and dimeric p85α. This can be explained in two ways: (1) The PTEN:p85α interaction is not a simple peptide:domain interaction between the PR2 peptide and the PTEN phosphatase domain, but also involves direct domain:domain interactions between dimeric p85α SH3-PR1-BH and PTEN. (2) The PTEN:p85α interaction is only a peptide:domain interaction between PR2 and PTEN, and the requirement for dimeric p85α is explained solely by a (flexible) cross-linking of two PTEN molecules. The combined SH3-BH-PR2:PTEN interaction from option 1 appears to be more suitable to explain all phenotypes we observe for the p85α association (protection from WWP2-mediated ubiquitination by outcompeting WWP2, enhanced membrane anchoring, increased phosphatase activity against soluble substrates). However, at this stage, we cannot categorically rule out option 2. We therefore first investigate the more likely option 1 and then evaluate option 2.

### (i) Orientation of PTEN with respect to p85α

In option 1, PTEN association with p85α has to be structurally compatible with the membrane association of PTEN. The surface of PTEN that binds to membranes and the catalytic site that requires binding phospholipid head groups are known (**Appendix figure 4A**). Based on this information, we oriented PTEN with respect to the membrane. We then used this PTEN: membrane interaction constraint to place p85α with respect to PTEN. In option 1, the requirement for a p85α dimer is justified by dimeric SH3-PR1-BH-PR2, providing a combined binding surface for PTEN. Given the symmetry of the BH domains and the distance between the PR2 regions, we expect that one PTEN binds to one p85α molecule of the homodimer and that two PTEN molecules bound to the p85α homodimer associate simultaneously with the membrane. Hence, the twofold symmetry axis of the BH dimer has to be perpendicular to the membrane plane. In a model where the p85α dimer is oriented with the SH3 domains towards the membrane, the SH3 domains, even if in close contact with the BH domains, would collide with the membrane and thus block simultaneous interaction of two bound PTEN molecules with the membrane (**Appendix figure 4B**). As a consequence, the p85α dimer is most likely oriented with the SH3 domains pointing away from the membrane. Such an orientation would also allow simultaneous binding of GTPases and PTEN to both the p85α dimer and the membrane.

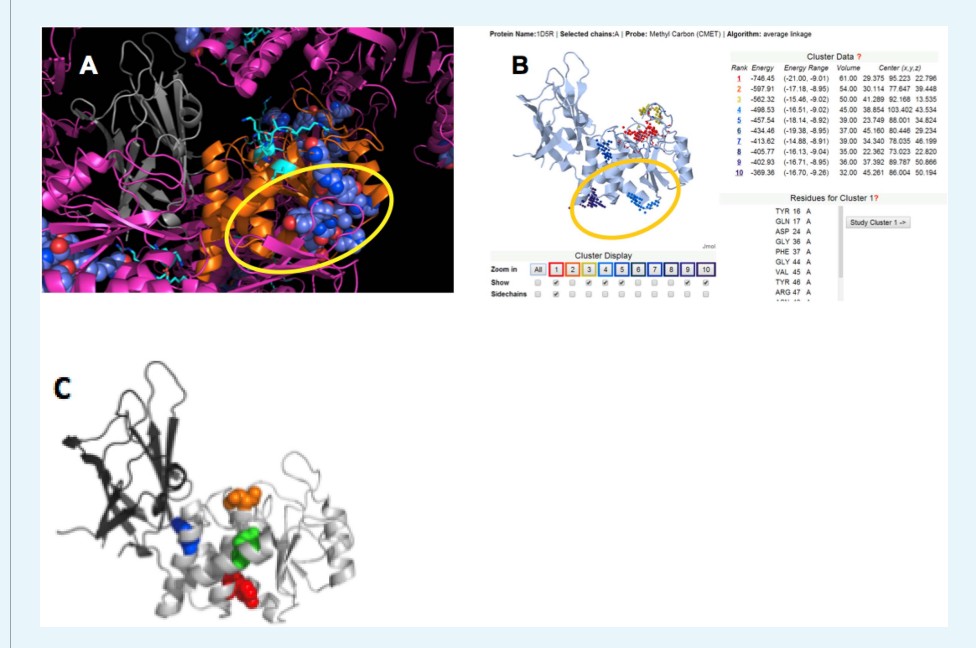

**Appendix figure 4**. (**A**) approximate positioning of PTEN with respect to the membrane (green). C2 residues required for membrane binding are highlighted. Orange spheres represent a bound L(+)-tartrate molecules found in the PTEN crystal structure (1D5R) and are thought to mimic interactions of PTEN with substrate phosphates(**Lee et al., 1999**). (**B**) When oriented with the SH3 domains (green) pointing towards the membrane, the two PTEN molecules cannot simultaneously interact with the membrane and with PR2. DOI: 10.7554/eLife.06866.025

## (ii) Identifying the most likely p85α binding site on PTEN

To identify the most likely p85α-binding site on PTEN, we applied four criteria: (1) protein–protein interaction sites are normally conserved across species; (2) protein–protein interaction sites are often implicated in crystal packing interactions; (3) PTEN mutations that affect binding to WWP2 should be close or overlapping with the p85α-interacting region on PTEN, because WWP2 and p85α compete for PTEN binding; and (4) a probable protein–ligand interaction site should obtain a significant score in ligand binding site prediction algorithms. *Appendix figure 5* shows that these four criteria highlight a similar region on the PTEN phosphatase domain, thus identifying a likely interaction site (encircled in *Appendix figure 5*).

**Appendix figure 5**. (**A**) PTEN is shown in the crystal lattice of 1D5R. Gray: C2; orange: phosphatase; magenta: symmetric molecules. Highly conserved regions are shown as blue spheres. (**B**) Result of Sitehound search (http://scbx.mssm.edu/sitehound/ (**Hernandez et al., 2009**) using a methyl carbon probe. Red and yellow dotted regions are close to the active site and hence unlikely BH interaction regions. (**C**) Maddika et al. have previously observed PTEN mutations that enhance (Y155F; green), slightly enhance (Y138F, red) and slightly decrease (Y174F; blue) WWP2 binding (**Maddika et al., 2011**). Y174 locates to the

C2:PTEN interface and is inaccessible to solvent and ligands. Y155 and Y138 both localize to the same side on PTEN and this side is the same as highlighted in (**A**) and (**B**).

## (iii) Selection of the most probable interaction model

We used the ClusPro server (http://cluspro.bu.edu/login.php) to produce an ensemble of molecular docking models of the BH:PTEN complex based on shape complementarity and interaction surface characteristics. This server was chosen because it produced the best results in the 2013 round of CAPRI (Critical Assessment of Predicted Interactions). Among the 30 best-scored solutions, five solutions clustered and were also compatible with all constraints mentioned above (*Appendix figure 6*), including the observation that the C2 domain does not contribute to binding (*Figure 3C*).

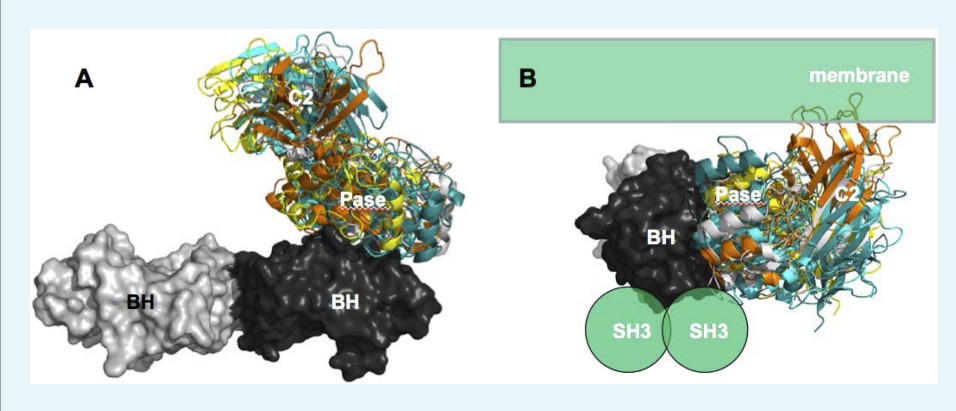

**Appendix figure 6**. Cluster of ClusPro models that satisfy all constraints. (**A**) Complexes viewed from the membrane; (**B**) side view with respect to the membrane.

In the docked position, the PTEN phosphatase domain is in proximity of the N-terminal helix–loop–helix fragment of the BH domain, and hence in proximity of both the BH:BH interface and the supposed location of the SH3:PR1 complex. We hypothesize that dimerization is required for either allosterically rearranging the PTEN binding site on the BH domain, or to allow stabilizing interactions between the second BH domain and the flexible 13 N-terminal residues of a PTEN molecule which binds with its phosphatase domain to the first BH domain. The PTEN:p85α interaction also involves PTEN R14, next to K13, a major PTEN ubiquitination site. To test this PTEN:p85α docking model, but also to test interaction option 1 (PTEN binds SH3-PR1-BH in addition to PR2) against option 2 (PTEN only binds PR2 region of p85α), we introduced the triple mutations I127A/I133A/E137A in the BH domain. The observations that this mutation on BH specifically affected PTEN binding, but not p85α dimerization, and that this mutation is synergistic with PR2 mutations support our molecular model (*Figure 6—figure supplement 1D*). More generally, the effect of these mutations supports interaction option 1. The observation that residues within this triple mutant affect PTEN binding eliminates the possibility that one PTEN phosphatase domain binds simultaneously to two BH domains within the dimer, because the dimensions of one PTEN phosphatase domain and the space constraints from the bound GTPase do not allow simultaneous contacts of one PTEN phosphatase domain with PR2, the three mutated residues and the two BH protomers.

## (iv) The suggested p85α-binding site of PTEN is highly mutated in cancer patients

Having observed that cancer patient-derived p85α mutations affect p85α dimerization (*Figure 5*), we asked whether patient-derived *PTEN* mutations would also map to the proposed p85α-binding site on PTEN. We therefore searched our in-house endometrial cancer dataset and the *Cancer* Genome Atlas (TCGA) for missense mutations in the PTEN phosphatase

domain and mapped these onto the structure model. Out of 62 mutations, 27 localized to the phosphatase surface that is oriented towards the membrane and contains the catalytic site (residues 24, 33–36, 39, 42, 45–47, 92, 93, 165, 167, 171, 123, 125–132, 136, 155, 159; red in *Figure 6—figure supplement 1D*); five localized to the C2:phosphatase interface (95, 170, 173, 175, 177; orange); 19 localized to the putative p85α-binding site proposed here (14, 15, 25, 27, 28, 30, 61, 64, 109, 112, 119, 138, 142–145, 148, 152, 153; green), and 11 localized to the opposite side (66, 71, 85, 87, 88, 96, 101, 105, 107, 108, 111; yellow). Hence up to 30% of missense *PTEN* mutations may localize to the putative p85α-binding site.

## Conclusions

From the data produced by our study and from previously published results, we compiled an ensemble of experimental, computational, structural and functional constraints for the PTEN: p85α interaction. These constraints can be summarized as (1) PTEN binds homodimeric but not monomeric p85α; (2) the dimeric BH:BH structure as observed in the crystal structure and supported by mutational analysis is maintained; (3) SH3:PR1 interactions occur in trans; (4) the PTEN:p85 complex is compatible with binding of Rab and other small GTPases to the p85α BH domain (*Figure 6—figure supplement 1A-B*); (5) p85α SH3 positioning is compatible with simultaneous GTPase and PTEN binding; (6) p85α interacts with the phosphatase domain of PTEN; (7) the PTEN C2 domain does not contribute to binding; (8) the PTEN:p85α interaction site overlaps with PTEN residues for binding WWP2; (9) p85 homodimers bind unphosphory-lated but not phosphorylated PTEN; (10) p85α-bound PTEN remains active, suggesting that the PTEN active sites can bind to membrane-bound PIP$_3$; (11) major PTEN ubiquitination sites are protected in the PTEN:p85α interface (K13 and/or K289) (*Trotman et al., 2007*; *Guo et al., 2012*); (12) p85α mutations at the p85α:PTEN interface weaken PTEN binding, but do not alter p85α homodimerization; and (13) the interface should appear biologically plausible, in terms of the surface covered and the amino acid composition. Finally, the p85α binding site on PTEN is most likely (14) a conserved region of PTEN; (15) a region predicted to be a ligand binding site; and (16) a region that is implicated in PTEN crystal packing interactions. We were able to develop a structural model that satisfies all these constraints (*Figure 6A*).

This 'hybrid' model (based on experimental and computational constraints) was corroborated by mutational analysis and correctly predicted cancer patient-derived p85α mutations that disrupt the p85α:PTEN-interacting interface (*Figures 5 and 6*). Our attempt to construct the p85α homodimer and p85α:PTEN structural models herein may therefore allow prediction of the biochemical phenotypes of mutations that occur along the interfaces in the cancer genomes. For a critical evaluation of this model, please see the discussion in the main text.

