## [Decision Letter]

Thank you for sending your work entitled “Regulation of the PI3K Pathway through a p85α Monomer-Homodimer Equilibrium” for consideration at *eLife*. Your article has been favorably evaluated by Tony Hunter (Senior Editor) and three reviewers, one of whom is a member of our Board of Reviewing Editors.

The Reviewing Editor and the other reviewers discussed their comments before we reached this decision, and the Reviewing editor has assembled the following comments to help you prepare a revised submission.

This is a carefully performed study that combines biochemical analysis and structural modeling to gain insight into the regulation of PTEN. A major conclusion of the study is the p85 homodimers compete for binding PTEN with the WWP2 E3 ligase. The conclusions of the study are, in general, consistent with the analytical data presented. However, in some instances the conclusions are over-stated and in other instances alternative interpretations of the data are not considered.

1) Figure 2. In the subsection headed “Disruption of p85α homodimerization is associated with decreased stability of PTEN through ubiquitination”, it is stated that WT p85 decreases and mutant p85 increases ubiquitination. This is true in a relative sense, but it appears that the extent of ubiquitination in each case is decreased compared with the LacZ control. These data should be presented more clearly.

2) The competition model is compelling, but it appears to be based on over-expression/transfection assays. Studies of endogenous proteins/loss-of-function analysis are needed to provide more convincing support for the authors' conclusions. For example, does knock-down of WWP2 increase the co-immunoprecipitation of p85 with PTEN (and the reverse experiment).

3) The interpretation of the transfection assays is limited to the deduction of direct binary interactions between PTEN with p85 homodimers and/or WWP2. Possible roles for signaling changes in the cells transfected with these reagents are not considered. For example, do changes in the PI3K pathway contribute to the changes in the PTEN interactions described? Feed-back or feed-forward signaling mechanisms may play a contributing role.

4) Are the biochemical interactions described for p85 in this study specific to p85α or does p85β similarly homodimerize and associate with PTEN through the indicated regions? The potential for redundancy in these mechanisms becomes important when one considers the influence of cancer-associated PIK3R1 mutations.

5) The structural model was constructed through a complicated procedure described in an eight page supplement, which describes some straightforward approaches (e.g. structural homology modeling and/or superposition) and others that are more tenuous (e.g. BUNCH with synthetic SAXS data). The resulting composite model depends on multiple assumptions, with some supporting biochemical/structural constraints, but is nevertheless speculative as the authors acknowledge. A triple mutation on a predicted PTEN-binding surface of the BH domain suggests that a least one of the mutated residues is located in the interface. However, there are no experiments that assess the structure in solution (e.g. actual SAXS experiments). It is difficult to assess the overall likelihood/uniqueness of the model or whether the constraints are sufficient to exclude alternative possibilities. In the absence of direct structural data for the relevant complex, the authors should be more circumspect with respect to the presentation of the speculative model and its implications, including discussion of potential caveats and possible alternative interpretations.

6) The *K*_*d*_ for dimerization of the SH3-PR1-BH-PR2 fragment of p85α is 10 fold lower than for the full length protein, which is a non-trivial difference. How do the authors explain this result?

---

## [Author Response]

*This is a carefully performed study that combines biochemical analysis and structural modeling to gain insight into the regulation of PTEN. A major conclusion of the study is the p85 homodimers compete for binding PTEN with the WWP2 E3 ligase. The conclusions of the study are, in general, consistent with the analytical data presented. However, in some instances the conclusions are over-stated and in other instances alternative interpretations of the data are not considered*.

*1)*
Figure 2*. In the subsection headed “Disruption of p85α homodimerization is associated with decreased stability of PTEN through ubiquitination”, it is stated that WT p85 decreases and mutant p85 increases ubiquitination. This is true in a relative sense, but it appears that the extent of ubiquitination in each case is decreased compared with the LacZ control. These data should be presented more clearly*.

The extent of PTEN ubiquitination in LacZ-transfected cells was higher than in p85α mutant-transfected cells because the amount of PTEN-bound p85α homodimer, which inhibits PTEN ubiquitination, was higher in mutant-transfected cells. The mutants, although they display decreased interaction with PTEN compared with wild-type p85α, were not completely defective in PTEN binding. We have now clarified this in the text (in the first paragraph of the subsection headed “Disruption of p85α homodimerization is associated with decreased stability of PTEN through ubiquitination”).

*2) The competition model is compelling, but it appears to be based on over-expression/transfection assays. Studies of endogenous proteins/loss-of-function analysis are needed to provide more convincing support for the authors' conclusions. For example, does knock-down of WWP2 increase the co-immunoprecipitation of p85 with PTEN (and the reverse experiment)*.

In response to the reviewers’ comments, we have performed loss-of function studies using siRNAs that efficiently target WWP-2 or p85α in HEC1A which is an endometrial cancer cell line expressing high levels of WWP-2 and p85α. The interaction between PTEN and p85α, or between PTEN and WWP-2 was increased after knockdown of WWP-2 or p85α respectively. The data are now presented in Figure 3—figure supplement 2 and in the text (in the subsection headed “p85α homodimer decreases PTEN binding to the E3 ligase WWP2”).

*3) The interpretation of the transfection assays is limited to the deduction of direct binary interactions between PTEN with p85 homodimers and/or WWP2. Possible roles for signaling changes in the cells transfected with these reagents are not considered. For example, do changes in the PI3K pathway contribute to the changes in the PTEN interactions described? Feed-back or feed-forward signaling mechanisms may play a contributing role*.

We agree with the reviewers that it would be interesting to examine whether the PTEN/p85 α/WWP2 interactions are subjected to changes in PI3K pathway signaling, especially through altering feedback and feedforward mechanisms. Negative feedback mediated by S6K (17) and positive feedback mediated by mTORC2 (19; 39) have been described in certain cell types and are considered to be the major feedback elements regulating the PI3K pathway. Example of feedforward signaling involves activation of mTOR, in which AKT phosphorylates mTOR directly or indirectly through inactivation of TSC1/TSC2 (15; 42).

To investigate whether these regulatory mechanisms could affect the interaction of PTEN with p85α or WWP2 when these molecules are altered by transfection, we suppressed S6K or mTORC2 signaling using S6K or Rictor siRNA, respectively, and rapamycin was used to inhibit mTORC1 activity in the cell line used in this study (KLE). In addition, we also targeted AKT activity using the specific AKT inhibitor MK2206. S6K siRNA increased AKT phosphorylation implicating the existence of negative feedback in KLE cells. In contrast to the suggested role in mediating a positive feedback at least in some cell lines, Rictor siRNA did not decrease phosphorylated AKT in the KLE cell line, although the siRNA decreased Rictor expression by 80%. Importantly, both S6K and Rictor siRNA had no effect on PTEN interactions with p85α or WWP2. Rapamycin and MK2206, which efficiently inhibited mTOR and AKT phosphorylation, respectively, also had no effect on these PTEN interactions with p85α or WWP. Thus, inhibition of these feedback and feedforward loops, and inhibition of different nodes of the PI3K pathway, does not affect the degree of PTEN association with p85α or WWP2, suggesting that the PTEN interactions with p85α or WWP2 are unlikely to be explained by mutant-triggered changes in PI3K downstream signaling or regulatory loops. These data and discussion are now added in Figure 2—figure supplement 1 and Figure 3—figure supplement 1 and in the text (in the subsections headed “Disruption of p85α homodimerization is associated with decreased stability of PTEN through ubiquitination” and “p85α homodimer decreases PTEN binding to the E3 ligase WWP2”).

*4) Are the biochemical interactions described for p85 in this study specific to p85α or does p85β similarly homodimerize and associate with PTEN through the indicated regions? The potential for redundancy in these mechanisms becomes important when one considers the influence of cancer-associated PIK3R1 mutations*.

The extent of functional redundancy between p85α and p85β is an interesting area of investigation. Harpur et al. have reported the homodimer formation of recombinant p85β proteins using size exclusion chromatography (16). To further assess the reviewers' question, we used immunoprecipitation assays and observed that p85β homodimerized and interacted with PTEN but the interactions were much weaker than observed for p85α. Further, in contrast to wild type p85α, p85β did not stabilize PTEN (data are now added in Figure 1—figure supplement 5).

The nSH2-iSH2-cSH2 fragment that binds p110 is 80% identical in sequence between p85α and p85β. In contrast, p85α and p85β share only 30% protein sequence identity between their BH domains that are critical for PTEN binding. Importantly, the BH region that mediates dimerization in p85α is not conserved in p85β, neither in sequence nor in structure (the structure of the p85β BH domain has been deposited in the PDB, accession number 2XS6, but no associated manuscript has been published to date). Accordingly, the BH domain dimer centered on M176 present in the p85α crystal is not present in the crystal lattice of p85β BH (see Figure 1—figure supplement 5). Instead, the p85β BH domain crystals contain another symmetric domain-domain interface. Therefore if the p85β BH domain contributes to stabilizing p85β homodimers, then it would do this using a different dimerization interface than p85α BH. Given that the dimeric structure of p85α BH domain is important for both PTEN binding and stabilization, these data explain our experimental observation of weak PTEN binding and lack of PTEN stabilization by p85β. Additional experiments, structure figure and discussion are now included in Figure 1—figure supplement 5 and in the text (in the subsection headed “The p85α BH:BH domain interaction contributes to stabilization of the p85α homodimer”).

*5) The structural model was constructed through a complicated procedure described in an eight page supplement, which describes some straightforward approaches (e.g. structural homology modeling and/or superposition) and others that are more tenuous (e.g. BUNCH with synthetic SAXS data). The resulting composite model depends on multiple assumptions, with some supporting biochemical/structural constraints, but is nevertheless speculative as the authors acknowledge. A triple mutation on a predicted PTEN-binding surface of the BH domain suggests that a least one of the mutated residues is located in the interface. However, there are no experiments that assess the structure in solution (e.g. actual SAXS experiments). It is difficult to assess the overall likelihood/uniqueness of the model or whether the constraints are sufficient to exclude alternative possibilities. In the absence of direct structural data for the relevant complex, the authors should be more circumspect with respect to the presentation of the speculative model and its implications, including discussion of potential caveats and possible alternative interpretations*.

Importantly, the hypothetical model presented is not used as a proof for our conclusions or indeed as a stand-alone model to explain the interactions between p85α monomers or between p85α and PTEN or WWP2, but to rather to guide and inform the experiments to be performed to elucidate the functional role of particular residues in p85α. We had emphasized this in the text and now clarify this further (e.g. in the Results, we state: “We used this structural information to build a theoretical homology model for the SH3:PR1 interaction to guide studies aimed at identifying the amino acids involved in the molecular interactions”. In the Discussion, we added the following: “while currently representing a useful working model for guiding and informing experimental analysis, our theoretical PTEN:p85α structure requires confirmation by an experimental structure”. Appendix 1 also provides further clarification on this point). Indeed, the predictive power and agreement of the model with the experimental mutation analysis support the utility of the molecular modeling.

Of course, in absence of an experimental 3D structural model, our complex structure remains hypothetical, and we cannot formally exclude other structural complexes that could satisfy our constraints, or that some of the constraints are wrong. As requested, we have now included a more critical discussion of the model and described its possible shortcomings (Discussion). To be more circumspect with respect to the presentation of the speculative model, we have limited its presentation in the manuscript from three to only one figure (the original Figure 6 have been reduced to the revised Figure 6). Additionally, to avoid risks of over-interpretation by the reader, we have now decreased the level of resolution of this figure, making it very schematic and cartoon-like in appearance.

*6) The* K_d_
*for dimerization of the SH3-PR1-BH-PR2 fragment of p85α is 10 fold lower than for the full length protein, which is a non-trivial difference. How do the authors explain this result*?

Indeed, the shorter N-terminal fragment consistently displayed a ten-fold lower dimerization *K*_*d*_ than the full-length protein. These differences are reproducible and significant. In AUC, these differences are also robust with respect to the model used for fitting, and the type of data acquisition (sedimentation equilibrium or velocity). Determination of the molecular basis of this observation would require complete experimental determination of structures of dimeric and monomeric p85 α (full-length and fragments thereof). Such structural analysis is non-trivial because of the size and flexibility of this multi-domain protein, and hence is beyond the scope of our manuscript.

Nonetheless, we speculate that the mildly inhibitory nature of the C-terminal nSH2-iSH2-cSH2 fragment arises from additional entropic penalty upon full-length dimerization, and/or from weak intramolecular interactions occurring between the SH3-BH fragment and the nSH2 -iSH2-cSH2 fragment in the monomer, which have to be displaced upon dimerization. This possibility has now been included in the Results section (“The difference in dimerization *K*_*d*_ between the N-terminal fragment and full-length p85α might indicate an additional entropic penalty arising upon full-length dimerization…”).